# Large-scale stochastic simulation of open quantum systems

**Aaron Sander** [1] ✉, **Maximilian Fröhlich** [2] ✉, **Martin Eigel** [2], **Jens Eisert** [3,4], **Patrick Gelß** [3,5], **Michael Hintermüller** [2], **Richard M. Milbradt** [1], **Robert Wille** [1,6,7] & **Christian B. Mendl** [1]

Understanding interactions between quantum systems and their environments is crucial for developing stable quantum technologies and accurate physical models. Yet, simulating open quantum systems with non-unitary dynamics remains computationally demanding. We introduce the tensor jump method (TJM), a scalable and embarrassingly parallel algorithm for stochastically simulating large-scale open quantum systems governed by Lindbladians. The TJM extends the Monte Carlo wave function (MCWF) approach to matrix product states, employs a dynamic time-dependent variational principle (TDVP) to minimize evolution errors, and introduces a sampling MPS to reduce timestep dependence. This method scales efficiently, ensuring convergence to Lindbladian dynamics independent of system size, as demonstrated both rigorously and numerically. We showcase its utility by simulating XXX Heisenberg models with up to a thousand spins on a consumer-grade CPU. The TJM represents a significant advance in open quantum system simulation, enabling exploration of dissipative many-body dynamics and the design of more stable quantum hardware.

Classical simulations are currently one of the most useful tools for comparing quantum theory with experimental results, as well as for providing benchmarks for the practical performance of quantum computers. Simulation results allow us to gain a deeper understanding of the underlying intricate physical mechanisms at work in complex quantum systems, as well as to build and scale more stable, reliable quantum computers and simulators. Quantum systems are notoriously difficult to simulate due to the exponential growth of parameters needed to represent larger systems. This growth leads to both increasing memory requirements to store exponentially many complex values and longer computational times to perform the operations needed to simulate their dynamics.

Tensor network methods, particularly *matrix product states* (MPS)[1–3], are famously at the forefront of classical tools for quantum simulation[4,5] as they facilitate the reduction of memory usage and

computational time compared to a full state vector representation, especially in the case of manageable entanglement growth[6–9]. Rather than representing quantum systems, i.e., quantum states, as state vectors that grow exponentially in dimension with system size, MPSs enable the system to be split into a tensor train[3] with each tensor representing a local system. This makes it possible to not only store the state of the quantum system with substantially fewer parameters, but also reduces many key components needed to simulate dynamics to local rather than global operations. This ansatz captures common correlation and entanglement patterns that present in quantum many-body systems well[2,10]. This property has led to the development of the *time-evolving block decimation* (TEBD) algorithm[11] which is a critical simulation method for MPS dynamics. The caveat is that for increasingly entangled states, especially those generated during time evolution, the bond dimension between these tensors grows, with the upper

[1]Technical University of Munich, Munich, Germany. [2]Weierstrass Institute for Applied Analysis and Stochastics, Berlin, Germany. [3]Freie Universität Berlin, Berlin, Germany. [4]Helmholtz-Zentrum Berlin für Materialien und Energie, Berlin, Germany. [5]Zuse Institute Berlin, Berlin, Germany. [6]Munich Quantum Software Company GmbH, Munich, Germany. [7]Software Competence Center Hagenberg GmbH (SCCH), Hagenberg, Austria. ✉e-mail: aaron.sander@tum.de; froehlich@wias-berlin.de

bound becoming exponential as the system gets larger. The ability to truncate the bonds to much lower dimensions has been proven to be a successful approximation method that enables TEBD and MPS to be reliable as an accurate simulation method for local Hamiltonians[12], even for highly-entangled states. However, truncation can break symmetries, violate energy conservation, and poorly approximate long-range interactions on fixed bond dimension manifolds. The *time-dependent variational principle* (TDVP)[13–16] addresses these issues by evolving quantum states in time through provably optimal MPS approximations with lower bond dimensions. TDVP thus has the advantage of preserving energy conservation and symmetries by avoiding TEBD's truncation errors, as well as not being confined to local interactions.

Substantial progress has been made in simulating quantum systems, such as finding ground states and closed system time evolution[1–3,6–16] and, more recently, quantum circuits[17–19]. However, simulating *open* quantum systems remains comparatively underexplored[20–24], even though environmental effects such as relaxation, dephasing, and thermal excitations are critical to understanding and mitigating noise in quantum technologies. Importantly, these interactions lead to unstable quantum computing architectures; in fact, quantum noise is the core antagonist in quantum technologies. In order to build better quantum devices, we need to precisely understand these noise processes to mitigate the error that they cause.

Lindblad master equations[25,26] (or *Lindbladians*) have long been the standard theoretical framework for modeling dissipative dynamics. This approach is the most general form of a completely positive and trace-preserving quantum dynamical semi-group. It evolves the density matrix $\rho(t) \in \mathbb{C}^{d^L \times d^L}$ directly, where $d$ is the physical dimension of one site and $L$ is the number of sites. Storing $\rho(t)$ as a dense matrix requires $\mathcal{O}(d^{2L})$ computer memory and becomes intractable with growing system size. An alternative is the *Monte Carlo wave function method* (MCWF)[27–32], which stochastically simulates individual quantum trajectories of pure states to approximate the corresponding density matrix. Although this approach partially circumvents the complexity of large density matrices, the computational cost still grows exponentially with system size as it relies on state vectors $|\Psi(t)\rangle \in \mathbb{C}^{d^L}$.

Early work toward scalable simulations of open quantum systems using tensor networks focused on *matrix product operator* (MPO) representations[20,21]. However, MPO-based approaches can suffer from positivity violations and numerical instabilities, especially when the resulting operators are not guaranteed to be positive semidefinite[33]. Alternative methods benefit from parallelization and lower bond dimensions than MPO approaches, though often at the cost of computational overhead[22]. A further distinction is typically made between approaches targeting full time evolution versus those computing steady states[34]. While steady states may require only modest bond dimensions[35], accurate simulation of non-equilibrium dynamics often leads to rapid bond growth.

In parallel, stochastic unravelings of open quantum systems, including both quantum jump and quantum state diffusion approaches, have been explored using tensor network representations[36–40]. These methods benefit from positivity by construction and natural parallelism across trajectories. Several recent works have successfully applied such techniques to electron-phonon dynamics[41] and photonic systems[42], employing Krylov and TDVP-based methods along with subspace truncation strategies.

Our work builds most directly on the foundational ideas of ref. 37, and we acknowledge the importance of more recent studies reviewed in refs. 43,44. However, existing trajectory-based methods often suffer from practical numerical instabilities, particularly when using TDVP to evolve under a non-Hermitian effective Hamiltonian. These issues can hinder scalability, despite theoretical advantage from tensor networks.

To overcome these challenges, we introduce the *tensor jump method* (TJM), a novel synthesis of quantum jump unravelings with *matrix product state* (MPS) techniques that achieves numerically stable and scalable simulation of large open systems. The core innovations are as follows:

- We introduce a *second-order Trotterization* of the non-Hermitian effective Hamiltonian, which decouples the Hermitian and dissipative parts of the evolution and eliminates discretization errors caused by this step. This formulation enables us to retrieve quantum states at intermediate times with reduced time-step sensitivity.
- We develop a *sampling MPS* that tracks the evolution under the reordered operator sequence, enabling extraction of trajectory states at arbitrary time steps without compromising the reduced Trotter error. This construction also allows the use of adaptive bond dimensions to efficiently represent long-time dynamics.
- We apply a *dynamic time-dependent variational principle* sweep to the Hermitian part of the system Hamiltonian, which switches between one-site and two-site TDVP based on the local bond dimension rather than truncating, while treating the dissipative evolution through an exact site-local contraction. This avoids instabilities that arise when applying TDVP to a non-Hermitian generator, maintains robustness across time steps, and swaps truncation error for projection error.

Unlike traditional t-DMRG or TEBD methods[45] which rely on fixed two-site gates and truncation heuristics, our approach ensures provably optimal evolution constrained to the variational manifold and supports rigorous convergence. While quantum trajectory methods using MPS have been explored previously, our method stands out by providing a principled resolution to some of the long-standing numerical bottlenecks, especially in regimes with strong dissipation or long simulation times.

At the foundational level, our approach leverages the stochastic *Monte Carlo wavefunction* (MCWF) framework[27–32], which is well suited to large-scale parallelization and has deep relevance for benchmarking quantum simulators. Beyond benchmarking, our method also opens avenues for classical dequantization in open-system regimes[46,47], extending the reach of classical simulation into parameter regimes previously believed to be intractable.

In summary, the TJM introduces a robust, convergent, and computationally efficient framework for simulating large-scale open quantum systems. While built upon established ingredients, its systematic integration of dynamic TDVP, higher-order unraveling, and stable sampling mechanisms positions it as a significant practical advance over prior trajectory-based MPS methods. These innovations fill critical gaps left by prior tensor network-based MCWF approaches, including ones using TDVP, enabling simulations at scale and with precision beyond the reach of earlier methods. Finally, we see this as a foundational work that can be further built on through new unraveling and splitting techniques[48–50] and the addition of non-Markovian processes[51]. An open-source implementation of this work can be found in the *MQT-YAQS* package available at ref. 52 as part of the *Munich Quantum Toolkit*[53].

## Results
### Simulation of open quantum systems
**Lindbladian master equations.** A large variety of processes in quantum physics can be captured by *quantum Markov processes*. On a formal level, they are described by the Lindblad master equation (or Lindbladian)[25,54]

$$\frac{d}{dt}\rho = -i[H_0, \rho] + \sum_{m=1}^{k} \gamma_m \left( L_m \rho L_m^\dagger - \frac{1}{2}\{L_m^\dagger L_m, \rho\} \right). \quad (1)$$

This gives rise to a dynamical semi-group that generalizes the Schrödinger equation. The quantum state $\rho$ and the Hamiltonian $H_0$ are Hermitian operators in the space of bounded operators $\mathcal{B}(\mathcal{H})$, acting on the Hilbert space of pure states of a quantum system $\mathcal{H}$. The first term $-i[H_0, \rho]$ corresponds to the unitary (closed/noise-free) time-evolution of $\rho$, where we have set $\hbar = 1$ to simplify notation. The summation captures the non-unitary dynamics of the system such that $\{L_m\}_{m=1}^k \subset \mathcal{B}(\mathcal{H})$ is a set of (non-unique) jump operators. These can be either Hermitian or non-Hermitian and correspond to noise processes in the system where $\{\gamma_m\}_{m=1}^k \subset \mathbb{R}_+$ is a set of coupling factors corresponding to the strength of each of these processes. These quantum jumps are defined as a sudden transition in the state of the system such as relaxation, dephasing, or excitation, which occur instantaneously, differentiating it from classical processes.

Directly computing the Lindbladian is one of the standard tools for exactly simulating open quantum systems[55]. However, its dependence on the dimension of the operator $\rho \in \mathbb{C}^{d^L \times d^L}$ limits the numerical computation of non-unitary dynamics to small system sizes. This highlights the need for substantially more scalable simulation methods of large-scale open quantum systems.

**Monte Carlo wave function method.** To decouple the simulation of open quantum systems from the scaling of the density matrix, the Lindbladian can be simulated stochastically using the *Monte Carlo wave function* method (MCWF)[27,29–32]. The MCWF breaks the dynamics of an open quantum system into a series of trajectories that represent possible evolutions of a time-dependent pure quantum state vector $|\Psi(t)\rangle \in \mathbb{C}^{d^L}$. Each trajectory corresponds to a non-unitary time evolution of the quantum state followed by stochastic application of quantum jumps, which leads to a sudden, non-continuous shift in its evolution. By averaging over many of these trajectories, the full non-unitary dynamics of the system can be approximated without directly solving the Lindbladian. More formally speaking, it gives rise to a stochastic process in projective Hilbert space that, on average, reflects the quantum dynamical semi-group.

Using the same jump operators as in Eq. (1), a non-Hermitian Hamiltonian can be constructed as

$$H = H_0 + H_D, \tag{2}$$

consisting of the system Hamiltonian $H_0$ and dissipative Hamiltonian defined as

$$H_D = -\frac{i}{2} \sum_{m=1}^k \gamma_m L_m^\dagger L_m. \tag{3}$$

Formally, this non-Hermitian Hamiltonian defines a time-evolution operator

$$U(\delta t) = e^{-iH\delta t}, \tag{4}$$

which can be used to evolve the state vector as

$$|\Psi^{(i)}(t + \delta t)\rangle = e^{-iH\delta t}|\Psi(t)\rangle, \tag{5}$$

where $H$ acts as an effective Hamiltonian in the time-dependent Schrödinger equation. The superscript ($i$) denotes the initial time-evolved state vector, which is not yet stochastically adjusted by the jump application process. Additionally, the dissipation caused by this operator does not preserve the norm of the state.

For the purpose of the MCWF derivation, the time evolution can be represented by the first-order approximation of the matrix exponential, where all $\mathcal{O}(\delta t^2)$ terms are dropped in the MCWF method[27]

$$\langle \Psi^{(i)}(t + \delta t)|\Psi^{(i)}(t + \delta t)\rangle = 1 - \delta p(t). \tag{6}$$

The denormalization $\delta p(t)$ can be used as a stochastic factor to determine if any jump has occurred in the given time step, where it can be seen as a summation of individual stochastic factors corresponding to the denormalization caused by each noise process

$$\delta p_m(t) = \delta t \gamma_m \langle \Psi(t)|L_m^\dagger L_m|\Psi(t)\rangle, \, m = 1, \ldots, k, \tag{7}$$

such that

$$\delta p(t) = \sum_{m=1}^k \delta p_m(t). \tag{8}$$

A random number $\epsilon$ is then sampled uniformly from the interval $[0, 1]$ and compared with the magnitude of $\delta p$. If $\epsilon \geq \delta p$, no jump occurs and the initial time-evolved state is normalized before moving onto the next time step

$$|\Psi(t + \delta t)\rangle = \frac{|\Psi^{(i)}(t + \delta t)\rangle}{\sqrt{\langle \Psi^{(i)}(t + \delta t)|\Psi^{(i)}(t + \delta t)\rangle}} = \frac{|\Psi^{(i)}(t + \delta t)\rangle}{\sqrt{1 - \delta p(t)}}. \tag{9}$$

If $\epsilon < \delta p$, a probability distribution of the possible jumps at the given time is created by

$$\Pi(t) = \{\Pi_1(t), \ldots, \Pi_k(t)\} \tag{10}$$

with the normalized stochastic factors $\Pi_m(t) = \delta p_m(t)/\delta p(t)$ such that $\sum_{m=1}^k \Pi_m(t) = 1$. A jump operator $L_m$ is then selected according to this probability distribution and applied directly to the pre-time-evolved quantum state

$$|\Psi(t + \delta t)\rangle = \frac{\sqrt{\gamma_m} L_m|\Psi(t)\rangle}{\sqrt{\langle \Psi(t)|L_m^\dagger \gamma_m L_m|\Psi(t)\rangle}} = \frac{\sqrt{\gamma_m} L_m|\Psi(t)\rangle}{\sqrt{\delta p_m(t)/\delta t}}, \tag{11}$$

simulating that the dissipative term has caused a jump during this time step. This methodology is then repeated until the desired elapsed time $T$ is reached, resulting in a single quantum trajectory and a final state vector $|\Psi(T)\rangle$. This process, which can be seen in Fig. 1b, gives rise to a stochastic process in projective Hilbert space called a piece-wise deterministic stochastic process in the limit $\delta t \to 0$.

This stochastic process is not only a classical Markovian stochastic process: In expectation, it exactly recovers the Markovian quantum process that is described by the dynamical semi-group generated by the master equation in Lindblad form

$$\rho(t) = \lim_{\delta t \to 0} \mathbb{E}\left(|\Psi_j(t)\rangle\langle\Psi_j(t)|\right). \tag{12}$$

For $N$ trajectories, the quantum state $\rho(t)$ at time $t$ is estimated as

$$\bar{\mu}(t) = \frac{1}{N} \sum_{j=1}^N |\Psi_j(t)\rangle\langle\Psi_j(t)|. \tag{13}$$

Rather than directly calculating the quantum state, the trajectories of state vectors can be stored and manipulated individually to calculate relevant expectation values. While this does not solve the exponential scaling with system size, it does provide a computational advantage over exactly calculating the quantum state in form of the density operator $\rho(t)$.

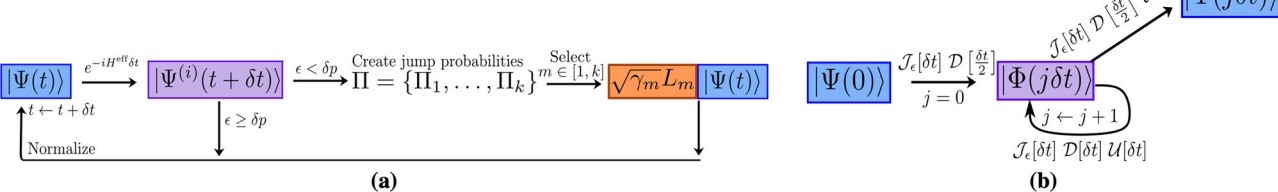

**Fig. 1 | Algorithms of the *Monte Carlo wave function* (MCWF) method and the *tensor jump method* (TJM). a** The MCWF algorithm stochastically simulates quantum trajectories by evolving a wavefunction $|\Psi(t)\rangle$ with a non-Hermitian effective Hamiltonian $e^{-iH^{\text{eff}}\delta t}$. A stochastic norm loss $\delta p = 1 - \langle\Psi^{(i)}(t+\delta t)|\Psi^{(i)}(t+\delta t)\rangle$ is compared against a uniform random number $\epsilon \in [0, 1]$. If $\epsilon \geq \delta p$, the result of the process is the dissipated state $|\Psi^{(i)}(t+\delta t)\rangle$. Otherwise, a quantum jump operator $L_j$ is applied based on computed jump probabilities to the pre-evolved state. The output of this stochastic process is then normalized. This process is repeated iteratively to simulate dissipation until the final time $T$. **b** The TJM extends this idea to tensor networks. A sampling MPS $|\Phi(j\delta t)\rangle$ is evolved by alternating applications of the dissipative sweep $\mathcal{D}$, a stochastic jump process $\mathcal{J}_\epsilon$, and dynamic TDVP $\mathcal{U}$. At any point, the quantum trajectory $|\Psi(j\delta t)\rangle$ can be sampled from $\Phi$ by applying a final sampling layer. The method retains the structure of MCWF while enabling low timestep error from second-order Trotterization without additional overhead. Further details of the operators $\mathcal{U}$, $\mathcal{D}$, and $\mathcal{J}_\epsilon$ are provided in section "Tensor jump method".

## Tensor jump method

While the MCWF improves the scalability of simulating open quantum systems in comparison to the exact Lindbladian computation, it still is bounded by the exponential scaling of state vectors. In conjunction with the success of MPS in simulating quantum systems, this provides an opportunity to extend the MCWF to a tensor network-based method which motivates this work. While directly solving the Lindbladian with MPOs has been explored[22,56], approaching this problem stochastically has only been examined for relatively small systems[36–42].

In this section, we introduce the components required for the *tensor jump method* (TJM). We first provide a general overview of the steps needed for the algorithm, without justification or explicit details, before walking the reader through the individual steps afterwards. We present the TJM with the aim to tackle the simulation of open quantum systems by designing every step to reduce the method's error and computational complexity as much as possible to maximize scalability.

**General mindset.** The general inspiration for the TJM is to transfer the MCWF to a highly efficient tensor network algorithm in which an MPS structure can be used to represent the trajectories, from which we can calculate the density operator or expectation values of observables. The stochastic time-evolution of one trajectory in the TJM consists of three main elements:

1. A dynamic TDVP $\mathcal{U}[\delta t]$.
2. A dissipative contraction $\mathcal{D}[\delta t]$.
3. A stochastic jump process $\mathcal{J}_\epsilon[\delta t]$.

The steps of the TJM described in this section are depicted in Fig. 1a and defined in sections "Dynamic TDVP", "Dissipative contraction", and "Stochastic jump process".

We begin with an initial state vector $|\Psi(0)\rangle$ at $t = 0$ encoded as an MPS. We wish to evolve this from some time $t \in [0, T]$ ($T = n\delta t(n > 0)$) according to a Hamiltonian $\boldsymbol{H_0}$, where we use bold font whenever the Hamiltonian is encoded as an MPO, along with some noise processes described by the set of jump operators $\{L_m\}_{m=1}^k$. Using the above elements, we can express the time-evolution operator $U(T)$ of one trajectory of the TJM as

$$U(T) = \prod_{i=0}^{n} \mathcal{F}_{n-i}[\delta t]. \tag{14}$$

which consists of $n$ subfunctions corresponding to each time step

$$\mathcal{F}_j[\delta t] = \begin{cases} \mathcal{J}_\epsilon[\delta t]\,\mathcal{D}\left[\frac{\delta t}{2}\right]\mathcal{U}[\delta t], & j = n, \\ \mathcal{J}_\epsilon[\delta t]\,\mathcal{D}[\delta t]\,\mathcal{U}[\delta t], & 0 < j < n, \\ \mathcal{J}_\epsilon[\delta t]\,\mathcal{D}\left[\frac{\delta t}{2}\right], & j = 0. \end{cases} \tag{15}$$

This set of subfunctions follows from higher-order Trotterization used to reduce the time step error (see section "Higher-order Trotterization"). However, this Trotterization causes the unitary evolution to lag behind the dissipative evolution by a half-time step, which is only corrected when the final operator $\mathcal{F}_n[\delta t]$ is applied.

To maintain the reduced time step error while being able to sample at each time step $t = 0, \delta t, 2\delta t, \ldots, T$ during a single simulation run, we introduce what we call a *sampling MPS* (denoted by $\boldsymbol{\Phi}$ while the quantum state itself is $\boldsymbol{\Psi}$.) This is initialized by the application of the first subfunction to the quantum state vector

$$|\boldsymbol{\Phi}(0)\rangle = \mathcal{F}_0[\delta t]|\boldsymbol{\Psi}(0)\rangle. \tag{16}$$

We use this to iterate through each successive time step

$$|\boldsymbol{\Phi}((j+1)\delta t)\rangle = \mathcal{F}_j[\delta t]|\boldsymbol{\Phi}(j\delta t)\rangle. \tag{17}$$

At any point during the evolution, we can retrieve the quantum state vector $|\boldsymbol{\Psi}(j\delta t)\rangle$ by applying the final function $\mathcal{F}_n$ to the sampling MPS as

$$|\boldsymbol{\Psi}(j\delta t)\rangle = \mathcal{F}_n[\delta t]|\boldsymbol{\Phi}(j\delta t)\rangle. \tag{18}$$

This allows us to sample at the desired time steps without compromising the reduction in time step error from applying the operators in this order.

We then repeat this time-evolution for $N$ trajectories from which we can reconstruct the density operator in MPO format $\boldsymbol{\rho}(t)$ according to Eq. (13)

$$\boldsymbol{\rho}(t) = \frac{1}{N}\sum_{i=1}^{N} |\boldsymbol{\Psi}_i(t)\rangle\langle\boldsymbol{\Psi}_i(t)|, \tag{19}$$

or, more conveniently, we can calculate the expectation values of observables $\boldsymbol{O}$ (stored as tensors or an MPO denoted by the bold font) for each individual trajectory

$$\langle\boldsymbol{O}(t)\rangle = \frac{1}{N}\sum_{i=1}^{N} \langle\boldsymbol{\Psi}_i(t)|\boldsymbol{O}|\boldsymbol{\Psi}_i(t)\rangle. \tag{20}$$

This results in an embarrassingly parallel process since each trajectory is independent and may be discarded after calculating the relevant expectation value.

**Higher-order Trotterization.** This section explains the steps to derive the subfunctions from Eq. (15) and provides our justification for doing so. We first define a generic non-Hermitian Hamiltonian created by the

system Hamiltonian $H_0$ and the dissipative Hamiltonian $H_D$ exactly as in Eq. (2)

$$H = H_0 + H_D. \tag{21}$$

From this, we create the non-unitary time-evolution operator that forms the basis of our simulation

$$U(\delta t) = e^{-i(H_0 + H_D)\delta t}. \tag{22}$$

In many quantum simulation use cases, including the derivation of the MCWF in Eq. (4), this operator would be split according to the first summands of the matrix exponential definition or Suzuki-Trotter decomposition[11,57–60]. However, higher-order splitting methods exist, which exhibit lower error[61–64]. While this comes at the cost of computational complexity, we show that in this case Strang splitting[64] (or second-order Trotter splitting) can be utilized to reduce the time step error from $\mathcal{O}(\delta t^2) \to \mathcal{O}(\delta t^3)$ with negligible change in computational time.

Applying Strang splitting, the time-evolution operator becomes

$$U^{(i)}(\delta t) = e^{-iH_D\frac{\delta t}{2}} e^{-iH_0\delta t} e^{-iH_D\frac{\delta t}{2}} + \mathcal{O}(\delta t^3), \tag{23}$$

where superscript $(i)$ denotes the initial time-evolution operator, which is not yet stochastically adjusted by the jump application process. These individual terms then define the dissipative operator

$$\mathcal{D}[\delta t] = e^{-iH_D\delta t} \tag{24}$$

and the unitary operator

$$\mathcal{U}[\delta t] = e^{-iH_0\delta t}. \tag{25}$$

For a time-evolution from $t \in [0, T]$ with terminal $T = n\delta t$ for $n$ time steps, the time-evolution operator takes the form

$$\begin{aligned} U^{(i)}(T) &= \left( \mathcal{D}\left[\tfrac{\delta t}{2}\right] \mathcal{U}[\delta t] \mathcal{D}\left[\tfrac{\delta t}{2}\right] \right)^n \\ &= \mathcal{D}\left[\tfrac{\delta t}{2}\right] \mathcal{U}[\delta t] \left( \mathcal{D}[\delta t] \mathcal{U}[\delta t] \right)^{n-1} \mathcal{D}\left[\tfrac{\delta t}{2}\right]. \end{aligned} \tag{26}$$

Here, we have combined neighboring half-time steps of dissipative operations, which is valid since $H_D$ commutes with itself for any choice of jump operators.

Note that this then takes the same form as the functions in Eq. (15) although without the stochastic operator $\mathcal{J}_\epsilon[j\delta t]$, leading to the initial time-evolution functions

$$\mathcal{F}_j^{(i)}[\delta t] = \begin{cases} \mathcal{D}\left[\tfrac{\delta t}{2}\right] \mathcal{U}[\delta t], & j = n, \\ \mathcal{D}[\delta t]\mathcal{U}[\delta t], & 0 < j < n, \\ \mathcal{D}\left[\tfrac{\delta t}{2}\right], & j = 0, \end{cases} \tag{27}$$

where

$$\mathcal{F}_j[\delta t] = \mathcal{J}_\epsilon[\delta t] \mathcal{F}_j^{(i)}[\delta t], \ \forall j. \tag{28}$$

**Dynamic TDVP.** For the unitary time-evolution $\mathcal{U}$, we employ a *dynamic TDVP* method. This is a combination of the *two-site TDVP* (2TDVP) and *one-site TDVP* (1TDVP) such that during the sweep, we allow the bond dimensions of the MPS to grow naturally up to a bond dimension $\chi_{\max}$, before confining the evolution to the current manifold, effectively capping the bond dimension and ensuring computational feasibility for the remainder of the simulation. More precisely, during each sweep, we locally use 2TDVP if the bond dimension has room to grow, otherwise we use 1TDVP at the site. Note that this means the maximum bond dimension is when the dynamic TDVP switches to

the local 1TDVP, rather than an absolute maximum seen in truncation, so some local bonds may be higher. This dynamic approach enables the necessary entanglement growth in the early stages while controlling the computational cost later on, making optimal use of available resources. In this section, we define a sweep as two half-sweeps, one from $\ell \in [1, \dots, L]$ and back for $\ell \in [L, \dots, 1]$. For a time step $\delta t$ this leads to two half-sweeps of $\frac{\delta t}{2}$.

To implement this approach, we express the state vector as a partitioning around one of its site tensors

$$|\mathbf{\Phi}\rangle = \left|\mathbf{\Phi}_{\ell-1}^L\right\rangle \otimes M_\ell \otimes \left|\mathbf{\Phi}_{\ell+1}^R\right\rangle. \tag{29}$$

By fixing the MPS to a mixed canonical form at site $\ell$ and applying the conjugate transpose of the partitioned single-site map $|\mathbf{\Phi}_{\ell-1}^L\rangle \otimes |\mathbf{\Phi}_{\ell+1}^R\rangle$ to each side of Eq. (83), the forward-evolving ODEs simplify to $L$ local ODEs of the form

$$\frac{d}{dt} M_\ell(t) = -i H_\ell^{\mathrm{eff}} M_\ell(t), \quad \ell = 1, \dots, L, \tag{30}$$

with a local effective Hamiltonian $H_\ell^{\mathrm{eff}}$, which dictates how to evolve each site tensor $M_\ell$ of the MPS. This process and the tensor network representation of $H_\ell^{\mathrm{eff}}$ is visualized in Fig. 2.

Once $H_\ell^{\mathrm{eff}}$ is computed, we matricize and exponentiate it using the Lanczos method[65]. After applying the Lanczos method, the site tensor is vectorized and updated according to the solution

$$M_\ell(t + \delta t) = e^{-iH_\ell^{\mathrm{eff}}\delta t} M_\ell(t). \tag{31}$$

A QR decomposition is then applied to shift the orthogonality center from $\ell \mapsto \ell + 1$ resulting in $M_\ell = \tilde{M}_\ell C_\ell$, where $\tilde{M}_\ell$ replaces the previous site tensor and creates an updated MPS $|\tilde{\mathbf{\Phi}}\rangle$. The bond tensor $C_\ell$ then evolves according to the simplified backward-evolving ODE, which is obtained by multiplying the conjugate transpose of $|\mathbf{\Phi}_\ell^L\rangle \otimes |\mathbf{\Phi}_{\ell+1}^R\rangle$ into each side of Eq. (84)

$$\frac{d}{dt} C_\ell(t) = +i\tilde{H}_\ell^{\mathrm{eff}} C_\ell(t), \quad \ell = 1, \dots, L-1, \tag{32}$$

with an effective Hamiltonian $\tilde{H}_\ell^{\mathrm{eff}}$ as shown in Fig. 2. Again, the Lanczos method is used to update the bond tensor

$$C_\ell(t + \delta t) = e^{+i\tilde{H}_\ell^{\mathrm{eff}}\delta t} C_\ell(t), \tag{32}$$

after which it is contracted along its bond dimension with the neighboring site $C_\ell(t + \delta t)M_{\ell+1}(t)$ to continue the sweep.

In the 2TDVP scheme, the process is modified by extending Eq. (77) to sum over two neighboring sites and adjusting Eq. (78) and Eq. (79) to act on both sites $\ell$ and $\ell + 1$

$$K_{\ell,\ell+1} = \left|\mathbf{\Phi}_{\ell-1}^L\right\rangle\left\langle\mathbf{\Phi}_{\ell-1}^L \otimes I_\ell \otimes I_{\ell+1} \otimes \left|\mathbf{\Phi}_{\ell+2}^R\right\rangle\right\rangle\left\langle\mathbf{\Phi}_{\ell+2}^R\right., \tag{33}$$

and

$$F_{\ell,\ell+1} = \left|\mathbf{\Phi}_{\ell-1}^L\right\rangle\left\langle\mathbf{\Phi}_{\ell-1}^L \otimes I_\ell \otimes \left|\mathbf{\Phi}_{\ell+1}^R\right\rangle\right\rangle\left\langle\mathbf{\Phi}_{\ell+1}^R\right.. \tag{34}$$

This results in the equations to update two tensors simultaneously

$$M_{\ell,\ell+1}(t + \delta t) = e^{-iH_{\ell,\ell+1}^{\mathrm{eff}}\delta t} M_{\ell,\ell+1}(t), \tag{35}$$

and

$$C_{\ell,\ell+1}(t + \delta t) = e^{+i\tilde{H}_{\ell,\ell+1}^{\mathrm{eff}}\delta t} C_{\ell,\ell+1}(t), \tag{36}$$

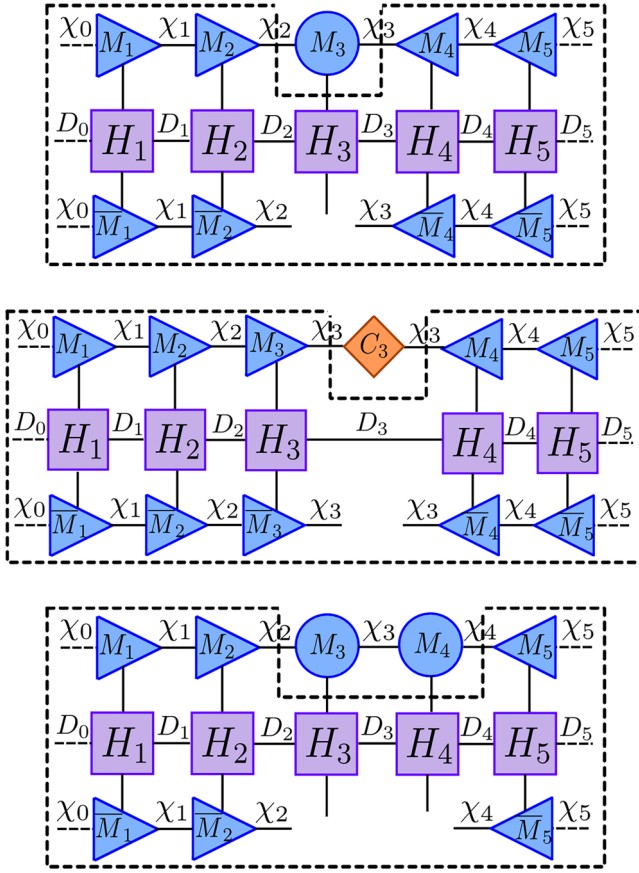

**Fig. 2 | This figure visualizes each step of the various forward- and backward-evolving equations of 1TDVP and 2TDVP indicated by the black dashed line.** This shows the reduced practical form of the network caused by the MPS's mixed canonical form. The tensors surrounded by the dashed lines (corresponding to $H^{\text{eff}}$) are contracted, exponentiated with the Lanczos method, then applied to the remaining tensors to update them. *Top* 1TDVP forward-evolving and 2TDVP backward-evolving network where $H_3^{\text{eff}}$ ($\bar{H}_{3,4}^{\text{eff}}$) is a degree-6 tensor. *Middle* 1TDVP backward-evolving network where $\bar{H}_3^{\text{eff}}$ is a degree-4 tensor. *Bottom* 2TDVP forward-evolving network where $H_{3,4}^{\text{eff}}$ is a degree-8 tensor.

where the effective Hamiltonians are defined as the tensor networks in Fig. 2. Note that the 1TDVP forward-evolution and the 2TDVP backward-evolution are functionally the same with a different prefactor in the exponentiation since $C_{\ell,\ell+1} = M_\ell$.

Compared to 1TDVP, this operation requires contraction of the bond between $M_\ell$ and $M_{\ell+1}$. After the time-evolution of the merged site tensors, a *singular value decomposition* (SVD) is applied with some threshold $s_{\max}$ such that the bond dimension $\chi_\ell$ can be updated and allowed to grow to maintain a low error.

Practically, this results in a DMRG-like[9] algorithm since TDVP reduces to a spatial sweep across sites for all $\ell = 1, ..., L$, where at each site (or pair of sites) we alternate between a forward-evolving update to a given site tensor, followed by a backward-evolving update to its bond tensor.

During the 1TDVP and 2TDVP sweeps, we compute the effective Hamiltonians using left and right environments which are updated and reused throughout the evolution[16]. Additionally, to effectively compute the matrix exponential, we apply the Lanczos method with a limited number of iterations[65], which significantly speeds up the computation of the exponential for large matrices, particularly when the bond dimensions grow large. Both of these are essential procedures to ensure that the TJM is scalable.

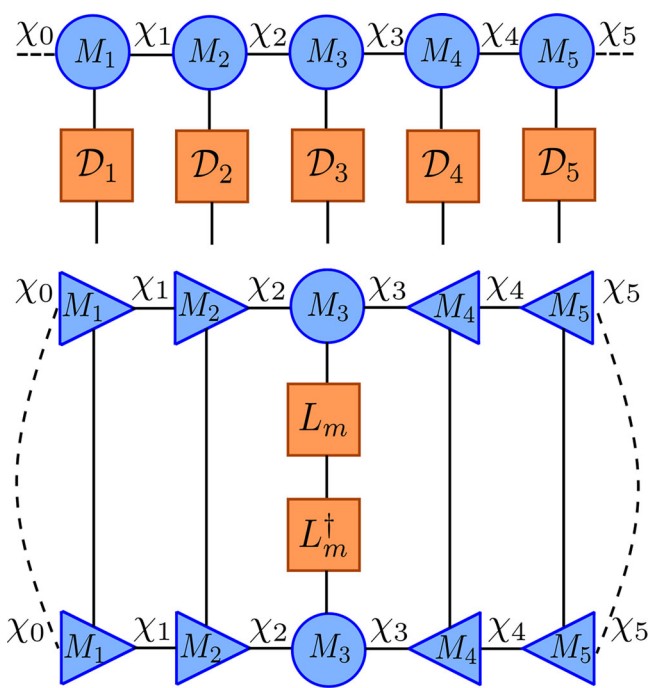

**Fig. 3 | Two components of the *tensor jump method* (TJM).** *Top* Illustration of the application of the factorized dissipative MPO $\mathcal{D}$ (constructed from local matrices) to an MPS $|\Psi\rangle$. Each local tensor corresponds to the exponentiation of local jump operators: $\mathcal{D}_\ell = e^{-\frac{1}{2}\delta t \sum_{j \in S(\ell)} L_j^{[\ell]\dagger} L_j^{[\ell]}}$. This operation does not change the bond dimension and does not require canonicalization. *Bottom* Visualization of the tensor network required to compute $\delta p_m$ for a given jump operator $L_m$, which corresponds to the expectation value $\langle L_m^\dagger L_m \rangle$. By putting the MPS in mixed canonical form centered at site $\ell$, contractions over tensors to the left and right reduce to identities. This enables efficient computation of the probability distribution $\Pi(t)$ via a sweep across the network, evaluating local jump probabilities $L_j$ at each site $j \in S(\ell)$.

To summarize, the described algorithm allows us to define the unitary time-evolution operator $\mathcal{U}[\delta t]$ as a piece-wise conditional with error in $\mathcal{O}(\delta t^3)$ since TDVP is a second-order method[64], given by

$$\mathcal{U}[\delta t] = \begin{cases} \prod_{\ell=1}^{L-2} \left( e^{-iH_{\ell,\ell+1}^{\text{eff}}\delta t} e^{+i\bar{H}_{\ell,\ell+1}^{\text{eff}}\delta t} \right) e^{-iH_{L-1,L}^{\text{eff}}\delta t} & : \chi_\ell < \chi_{\max}, \\ \prod_{\ell=1}^{L-1} \left( e^{-iH_\ell^{\text{eff}}\delta t} e^{+i\bar{H}_\ell^{\text{eff}}\delta t} \right) e^{-iH_L^{\text{eff}}\delta t} & : \chi_\ell \geq \chi_{\max}. \end{cases} \tag{37}$$

**Dissipative contraction.** The dissipative operator $\mathcal{D}[\delta t]$ is created by exploiting the structure of the exponentiation of local jump operators in Eq. (24). This operation can be factorized into purely local operations due to the commutativity of the sums of single site operators. As a result, the dissipation term is equivalent to a single contraction of the dissipative operator $\mathcal{D}[\delta t]$ into the current MPS $|\Phi(t)\rangle$. Additionally, this does not increase the bond dimension when applied to an MPS and the dissipative contraction is exact *without inducing errors*. This contraction is visualized in Fig. 3.

In this work, we focus on single-site jump operators $L_m \in \mathbb{C}^{d^L \times d^L}$ where each $L_m$ is a tensor product of identity matrices $I \in \mathbb{C}^{d \times d}$ and a local non-identity operator $L_m^{[\ell]} \in \mathbb{C}^{d \times d}$ acting on some site $\ell = 1, ..., L$. Specifically, for each $m = 1, ..., k$, the operator $L_m$ can be written as

$$L_m = I^{\otimes(\ell-1)} \otimes L_m^{[\ell]} \otimes I^{\otimes(L-\ell+1)} = I_{\backslash \ell} \otimes L_m^{[\ell]}, \tag{38}$$

where $L_m^{[\ell]}$ acts on the $\ell^{\text{th}}$ site and $I_\ell$ denotes the identity operator acting on all sites except $\ell$. This allows the dissipative Hamiltonian to be

localized site-wise

$$H_D = -\frac{i}{2}\sum_{m=1}^{k}\gamma_m L_m^\dagger L_m$$
$$= -\frac{i}{2}\sum_{\ell=1}^{L}\left[\sum_{j\in S(\ell)}\gamma_j(I_{\backslash\ell}\otimes L_j^{[\ell]\dagger}L_j^{[\ell]})\right], \quad (39)$$

where $S(\ell)\subseteq[1,...,k]$ is the set of indices for the jump operators in $\{L_m\}_{m=1}^{k}$ for which there is a non-identity term at site $\ell$. When exponentiated, this results in

$$\mathcal{D}[\delta t] = e^{-i\left(-\frac{i}{2}\sum_{\ell=1}^{L}\left[\sum_{j\in S(\ell)}\gamma_j\left(I_{\backslash\ell}\otimes L_j^{[\ell]\dagger}L_j^{[\ell]}\right)\right]\right)\delta t}$$
$$= \prod_{\ell=1}^{L}e^{-\frac{1}{2}\sum_{j\in S(\ell)}\gamma_j(I_{\backslash\ell}\otimes L_j^{[\ell]\dagger}L_j^{[\ell]})\delta t}$$
$$= \prod_{\ell=1}^{L}e^{I_{\backslash\ell}\otimes(-\frac{1}{2}\delta t\sum_{j\in S(\ell)}\gamma_j L_j^{[\ell]\dagger}L_j^{[\ell]})} \quad (40)$$
$$= \bigotimes_{\ell=1}^{L}e^{-\frac{1}{2}\delta t\sum_{j\in S(\ell)}\gamma_j L_j^{[\ell]\dagger}L_j^{[\ell]}} = \bigotimes_{\ell=1}^{L}\mathcal{D}_\ell[\delta t].$$

The resulting operator corresponds to a factorization of site tensors where $\mathcal{D}_\ell[\delta t]\in\mathbb{C}^{d\times d}$ for $\ell\in[1,...,L]$.

**Stochastic jump process.** Following each $\mathcal{F}_j^{(i)}$, we perform the jump process $\mathcal{J}_\epsilon[\delta t]$ for determining if (and how) jump operators should be applied to the MPS. $\mathcal{J}_\epsilon[\delta t]$ is the value of a stochastic function $\mathcal{J}$ that maps a randomly selected $\epsilon\in[0,1]$ in combination with a time step size $\delta t$ to an operator. This operator is either the identity operator if no jump occurs or a single site-jump operator $L_m^{[\ell]}, m=1,\ldots,k$ in the case of a jump,. This operator, encoded as a single-site tensor, is then contracted into the sampling MPS $|\mathbf{\Phi}(t)\rangle$.

We apply the jump process $\mathcal{J}_\epsilon[\delta t]$ following each operation defined in Eq. (15). This means that first the initial time-evolved state vector from $t\mapsto t+\delta t$ is created

$$|\mathbf{\Phi}^{(i)}(t+\delta t)\rangle = \mathcal{F}_j^{(i)}[\delta t]|\mathbf{\Phi}(t)\rangle.$$

From this we create

$$|\mathbf{\Phi}(t+\delta t)\rangle = \mathcal{J}_\epsilon[\delta t]|\mathbf{\Phi}^{(i)}(t+\delta t)\rangle, \quad (41)$$

where the operator $\mathcal{J}_\epsilon[\delta t]$ acts as described in the following.

First, the overall stochastic factor $\delta p(t)$ is determined as in the MCWF by taking the inner product of this state, where we begin to sweep across the state to maintain a mixed canonical form following the dissipative contraction. This reduces the calculation to contracting the final tensor of the MPS with itself, i.e.,

$$\delta p = 1 - \langle\mathbf{\Phi}^{(i)}(t+\delta t)|\mathbf{\Phi}^{(i)}(t+\delta t)\rangle$$
$$= 1 - \sum_{\sigma_L=1}^{d}\sum_{a_{L-1},a_L=1}^{\chi_{L-1},\chi_L}M_L^{\sigma_L,a_{L-1},a_L}\overline{M}_L^{\sigma_L,a_{L-1},a_L}. \quad (42)$$

In contrast to the MCWF, we do not use a first-order approximation of $e^{-iH\delta t}$ to calculate $\delta p(t)$ since the time-evolution has been carried out by the TDVP projectors and the dissipative contraction. Next, $\epsilon\in[0,1]$ is sampled uniformly, which subsequently leads to two possible paths.

If $\epsilon\geq\delta p$, we normalize $|\mathbf{\Phi}^{(i)}(t+\delta t)\rangle$. In this case, the dissipative contraction itself represents the noisy interactions from time $t\mapsto t+\delta t$.

If $\epsilon<\delta p$, we generate the probability distribution of all possible jump operators using the initial time-evolved state

$$\Pi_m(t) = \frac{\delta t\gamma_m}{\delta p(t)}\langle\mathbf{\Phi}^{(i)}(t+\delta t)L_m^\dagger L_m|\mathbf{\Phi}^{(i)}(t+\delta t)\rangle \quad (43)$$

for $m=1,...,k$. At any given site $\ell$, we can calculate the probability $\Pi_j(t)$ for $j\in S(\ell)$ according to $\langle\mathbf{\Phi}^{(i)}(t+\delta t)L_j^\dagger L_j|\mathbf{\Phi}^{(i)}(t+\delta t)\rangle$ for the relevant jump operators $L_j$. When performed in a half-sweep across $\ell=[1,...,L]$ where at each site the MPS is fixed into its mixed canonical form, the probability is calculated by a contraction of the jump operator and the site tensor $M_\ell$

$$N_\ell^{\sigma'_\ell,a_\ell,a_{\ell-1}} = \sum_{\sigma_\ell=1}^{d}L_m^{\sigma'_\ell,\sigma_\ell}M_\ell^{\sigma_\ell,a_{\ell-1},a_\ell}. \quad (44)$$

Note that this is not directly updating the MPS but rather using the current state of its tensors to calculate the stochastic factors. Then, the inner product of this new tensor with itself is evaluated while scaling it accordingly as done in the MCWF

$$\Pi_m(t) = \frac{\delta t\gamma_m}{\delta p(t)}\sum_{\sigma'_\ell=1}^{d}\sum_{a_{\ell-1},a_\ell=1}^{\chi_{\ell-1},\chi_\ell}N_\ell^{\sigma'_\ell,a_{\ell-1},a_\ell}\overline{N}_\ell^{\sigma'_\ell,a_\ell,a_{\ell-1}}. \quad (45)$$

This is repeated for $j\in S(\ell)$ until all jump probabilities at site $\ell$ are calculated. We then move to the next site $\ell\mapsto\ell+1$ performing the same process until $\ell=L$ and the half-sweep is complete.

This yields the probability distribution $\Pi(t)=\{\Pi_m(t)\}_{m=1}^{k}$ from which we can randomly select a jump operator $L_m$ to apply to $|\mathbf{\Phi}^{(i)}(t+\delta t)\rangle$. This is achieved by multiplying $L_m$ into the relevant site tensor $M_\ell$ with elements

$$\tilde{M}_\ell^{\sigma_\ell,\alpha_\ell,\alpha_{\ell-1}} := \sqrt{\gamma_m}\sum_{\sigma'_\ell=1}^{d}L_m^{[\ell]\sigma'_\ell,\sigma_\ell}M_\ell^{\sigma'_\ell,\alpha_\ell,\alpha_{\ell-1}}.$$

The result is the updated MPS

$$|\mathbf{\Phi}(t+\delta t)\rangle = \sum_{\sigma_1,...,\sigma_L=1}^{d}M_1^{\sigma_1}\ldots\tilde{M}_\ell^{\sigma_\ell}\ldots M_L^{\sigma_L}|\sigma_1,\ldots,\sigma_L\rangle.$$

The state is then normalized through successive SVDs before moving onto the next time step. This allows the state to naturally compress as the application of jumps often suppress entanglement growth in the system. Note that this is a fundamental departure from the MCWF in which the jump is applied to the state at the previous time $t$.

**Algorithm.** With all necessary tensor network methods established, we can now combine the procedures from the previous sections to construct the complete TJM algorithm.

The TJM requires the following components:
1. $|\mathbf{\Psi}(0)\rangle$: Initial quantum state vector, represented as an MPS.
2. $H_0$: Hermitian system Hamiltonian, represented as an MPO.
3. $\{L_m\}_{m=1}^{k}, \{\gamma_m\}_{m=1}^{k}$: A set of single-site jump operators stored as matrices with their respective coupling factors.
4. $\delta t$: Time step size.
5. $T$: Total evolution time.
6. $\chi_{max}$: Maximum allowed bond dimension.
7. $N$: Number of trajectories.

Once these components are defined, the noisy time evolution from $t\in[0,T]$ is performed by iterating through each time step using

the operators described in Eq. (15). We first initialize the sampling MPS for the time evolution. The initial state $|\mathbf{\Psi}(0)\rangle$ is evolved using the operator $\mathcal{F}_0 = \mathcal{J}_\epsilon[\delta t]\mathcal{D}[\frac{\delta t}{2}]$, which includes a half-time step dissipative contraction and a stochastic jump process $\mathcal{J}_\epsilon[\delta t]$

$$|\mathbf{\Phi}(\delta t)\rangle = \mathcal{F}_0|\mathbf{\Psi}(0)\rangle. \tag{46}$$

The algorithm then evolves the system to each successive time step $t = j\delta t$ using the operators $\mathcal{F}_j = \mathcal{J}_\epsilon[\delta t]\mathcal{D}[\delta t]\mathcal{U}[\delta t]$ as seen in Eq. (18)

$$|\mathbf{\Phi}((j+1)\delta t)\rangle = \mathcal{F}_j|\mathbf{\Phi}(j\delta t)\rangle. \tag{47}$$

Each iteration involves the following.
- A full time step unitary operation $\mathcal{U}[\delta t]$ using the dynamic TDVP:
    - If any bond dimension $\chi_\ell < \chi_{\max}$, 2TDVP is applied, allowing the bond dimensions to grow dynamically.
    - If the bond dimension has reached $\chi_{\max}$, 1TDVP is applied to constrain further growth and maintain computational efficiency.
- A full time step dissipative contraction $\mathcal{D}[\delta t]$ where the non-unitary part of the evolution is applied.
- A jump process to determine whether quantum jumps occur, including normalization of the state.

This process is repeated until the time evolution reaches terminal $T$.

At any point during the time evolution, the quantum state can be retrieved by applying the final function $\mathcal{F}_n = \mathcal{J}_\epsilon[\delta t]\mathcal{D}[\frac{\delta t}{2}]\mathcal{U}[\delta t]$ as if the system has evolved to the stopping time

$$|\mathbf{\Psi}(\delta t)\rangle = \mathcal{F}_n|\mathbf{\Phi}(\delta t)\rangle. \tag{48}$$

The state is then normalized. This allows for the state to be inspected or observables to be calculated at any intermediate time without compromising the reduction in error from the Strang splitting. The above process is repeated from the beginning for each of the $N$ trajectories, providing access to a compact storage of the density matrix and its evolution as well as the ability to calculate expectation values.

## Computational complexity and convergence guarantees

While the TJM method proposed here is in practice highly functional and performs well, it can also be equipped with tight bounds concerning the computational and memory complexity as well as with convergence guarantees. This section is devoted to justifying this utility in approximating Lindbladian dynamics by analyzing the mathematical behavior of the TJM based on its convergence and error bounds. Since we assert that the TJM is highly-scalable compared to other methods, this analytical proof serves to lend credence to large-scale results, which may have no other method against which we can benchmark. These important points are discussed here, while substantial additional details are presented in the methods and appendix sections.

**Computational effort.** We derive and compare the computational and memory complexity of the exact calculation of the Lindblad equation, the MCWF, a Lindblad MPO, and the TJM method in detail in sections "Computational effort of running the simulation", "Resources required for storing the results", "Resources required for calculating expectation values" and Appendix 3. The results are summarized in Supplementary Table 1 in Appendix 1, showcasing the highly beneficial and favorable scaling of the computational and memory complexities of the TJM method.

**Monte Carlo convergence.** The convergence of the TJM is stated in terms of the density matrix standard deviation, which we define as follows.

**Definition 1.** (Density matrix variance). Let $\|\cdot\|$ be a matrix norm, and let $X \in \mathbb{C}^{n\times n}$ be a matrix-valued random variable defined on a probability space $(\Omega, \mathcal{F}, \mathbb{P})$, where $\mathbb{P}$ is a probability measure. The variance of $X$ with respect to the norm $\|\cdot\|$ is defined as

$$\mathbb{V}[X] = \mathbb{E}\left[\|X - \mathbb{E}[X]\|^2\right], \tag{49}$$

where $\mathbb{E}[X]$ denotes the expectation of $X$. The expectation $\mathbb{E}[X]$ is computed entrywise, with each entry being the expectation according to the respective marginal distributions of the entries. Specifically, for each $i, j$,

$$\mathbb{E}[X]_{i,j} = \mathbb{E}_{\mathbb{P}_{i,j}}[x_{i,j}], \tag{50}$$

where $x_{i,j}$ is the $(i, j)$-th entry of the matrix $X$, and $\mathbb{P}_{i,j}$ is the marginal distribution of $x_{i,j}$ induced by $\mathbb{P}$. The expectation value of the norm of a matrix $\mathbb{E}[\|\cdot\|]$ is defined as the multidimensional integral over the function $\|\cdot\|: \mathbb{C}^{n\times n} \to \mathbb{R}$ according to its marginal distributions $\mathbb{P}_{i,j}$.

The standard deviation of $X$ with respect to the norm $\|\cdot\|$ is then defined as

$$\sigma(X) = \sqrt{\mathbb{V}[X]} = \sqrt{\mathbb{E}\left[\|X - \mathbb{E}[X]\|^2\right]}. \tag{51}$$

For the proof of the convergence, we furthermore need additional properties of the density matrix variance, which specifically hold true for the Frobenius norm. They are given in Appendix 2. With this, the convergence of TJM can be proved as follows.

**Theorem 2.** (Convergence of TJM). Let $d \in \mathbb{N}$ be the physical dimension and $L \in \mathbb{N}$ be the number of sites in the open quantum system described by the Lindblad master equation Eq. (1). Furthermore, let $\boldsymbol{\rho}_N(t) = \frac{1}{N}\sum_{i=1}^{N}|\mathbf{\Psi}_i(t)\rangle\langle\mathbf{\Psi}_i(t)|$ be the approximation of the solution $\rho(t)$ of the Lindblad master equation in MPO format at time $t \in [0, T]$ for some ending time $T > 0$ and $N \in \mathbb{N}$ trajectories, where $|\mathbf{\Psi}_i(t)\rangle$ is a trajectory sampled according to the TJM in MPS format of full bond dimension. Then, the expectation value of the approximation of the corresponding density matrix $\rho_N(t) \in \mathbb{C}^{d^L \times d^L}$ is given by $\rho(t)$ and there exists a $c > 0$ such that the standard deviation of $\rho_N(t)$ can be upper bounded by

$$\sigma(\rho_N(t)) \le \frac{c}{\sqrt{N}} \tag{52}$$

for all matrix norms $\|\cdot\|$ defined on $\mathbb{C}^{d^L, d^L}$.

The full proof of Theorem 2 can be found in Appendix 2. For the convenience of the reader, a short sketch of the proof is provided here:

**Proof.** By the law of large numbers and the equivalence proof in Appendix 1, it follows that for every $N \in \mathbb{N}$ and every time $t \in [0, T]$ we have that $\mathbb{E}[\rho_N(t)] = \rho(t)$. The proof is carried out in state vector and density matrix format since MPSs and MPOs with full bond dimension exactly represent the corresponding vectors and matrices. Thus, we denote the state vector of a trajectory sampled according to the TJM at time $t$ by $|\Psi_i(t)\rangle$. Using the definition of the variance in the Frobenius norm and the fact that each trajectory is independently and identically distributed, we see that the variance of $\mathbb{V}_F[\rho_N(t)]$ decreases linearly

with $N$ by

$$
\begin{aligned}
\mathbb{V}_F[\rho_N(t)] &= \mathbb{V}_F\left[\frac{1}{N}\sum_{i=1}^{N}|\Psi_i(t)\rangle\langle\Psi_i(t)|\right] \\
&= \frac{1}{N^2}\mathbb{V}_F\left[\sum_{i=1}^{N}|\Psi_i(t)\rangle\langle\Psi_i(t)|\right]
\end{aligned}
\tag{53}
$$

$$
= \frac{1}{N^2}\sum_{i=1}^{N}\mathbb{V}_F[|\Psi_i(t)\rangle\langle\Psi_i(t)|]
\tag{54}
$$

$$
= \frac{1}{N}\mathbb{V}_F[|\Psi_1(t)\rangle\langle\Psi_1(t)|] \le \frac{4}{N},
\tag{55}
$$

where the second to last step follows from the identical distribution of all $|\Psi_i(t)\rangle\langle\Psi_i(t)|$ for $i = 1, \ldots, N$. Hence, the Frobenius norm standard deviation is upper bounded by

$$
\sigma_F[\rho_N(t)] = \frac{1}{\sqrt{N}}\sigma_F[|\Psi_1(t)\rangle\langle\Psi_1(t)|] \le \frac{2}{\sqrt{N}}.
\tag{56}
$$

By the equivalence of norms on finite vector spaces, there exists $c_1, c_2 \in \mathbb{R}$ such that $c_1\|A\|_F \le \|A\| \le c_2\|A\|_F$ for all complex square matrices $A$ and all matrix norms $\|\cdot\|$. Consequently, the convergence rate $\mathcal{O}(1/\sqrt{N})$ also holds true in trace norm and any other relevant matrix norm and is independent of system size. The statement then follows directly. □

**Error measures.** The major error sources of the TJM are as follows:
1. The time step error of the *Strang splitting* ($\mathcal{O}(\delta t^3)$)[62],
2. The time step error of the dynamic TDVP ($\mathcal{O}(\delta t^3)$ per time step and $\mathcal{O}(\delta t^2)$ for the whole time-evolution), and
3. The *projection error* of the dynamic TDVP.

Note that for 2TDVP the projection error is exactly zero if we consider Hamiltonians with only nearest neighbor interactions[15,16] such that the projection error depends on the Hamiltonian structure. If each of the mentioned errors were zero, we would in fact calculate the MCWF from which we know that its stochastic uncertainty decreases with increasing number of trajectories according to the standard Monte Carlo convergence rate as shown in section "Monte Carlo convergence".

The projection error of 1TDVP can be calculated as the norm of the difference between the true time evolution vector $\boldsymbol{H_0}|\boldsymbol{\Phi}\rangle$ and the projected time evolution vector $P_{\mathcal{M}_{\chi,|\boldsymbol{\Phi}\rangle}}\boldsymbol{H_0}|\boldsymbol{\Phi}\rangle$. It depends on the structure of the Hamiltonian and the chosen bond dimensions $\chi \in \mathbb{N}^{L+1}$

$$
\epsilon(\chi) = \|(I - P_{\mathcal{M}_{\chi,|\boldsymbol{\Phi}\rangle}})\boldsymbol{H_0}|\boldsymbol{\Phi}\rangle\|_2.
\tag{57}
$$

This error can be evaluated as shown in ref. 66. It is well-known that the 1TDVP projector solves the minimization problem

$$
P_{\mathcal{M}_{\chi,|\boldsymbol{\Phi}\rangle}}\boldsymbol{H_0}|\boldsymbol{\Phi}\rangle = \underset{M\in\mathcal{M}_\chi}{\operatorname{argmin}} \|\boldsymbol{H_0}|\boldsymbol{\Phi}\rangle - M\|_2.
\tag{58}
$$

It can thus be noted that TJM uses the computational resources in an optimal way regarding the accuracy in time-evolution[13]. The errors in the dissipative contraction and in the jump application are both zero.

## Benchmarking

To benchmark the proposed TJM, we consider a 10-site *transverse-field Ising model* (TFIM),

$$
H_0 = -J\sum_{i=1}^{L-1}Z^{[i]}Z^{[i+1]} - g\sum_{j=1}^{L}X^{[j]},
\tag{59}
$$

at the critical point $J = g = 1$ where $X^{[i]}$ and $Z^{[i]}$ are Pauli operators acting on the $i$-th site of a 1D chain. We evolve the state $|0\ldots0\rangle$ according to this Hamiltonian under a noise model that consists of single-site relaxation and dephasing operators,

$$
\sigma_- = \begin{pmatrix} 0 & 1 \\ 0 & 0 \end{pmatrix}, \quad Z = \begin{pmatrix} 1 & 0 \\ 0 & -1 \end{pmatrix},
\tag{60}
$$

on *all* sites in the lattice. Thus, our set of Lindblad jump operators is given by

$$
\{L_m\}_{m=1}^{2L} = \{\sigma_-^{[1]}, \ldots, \sigma_-^{[L]}, Z^{[1]}, \ldots, Z^{[L]}\}
\tag{61}
$$

with coupling factors $\gamma = \gamma_- = \gamma_z = 0.1$.

All simulations reported here (other than the comparison with the 100-site steady state, which was run on a server) were performed on a consumer-grade Intel i5-13600KF CPU (5.1 GHz, 14 cores, 20 threads), using a parallelization scheme in which each TJM trajectory runs on a separate thread. This setup exemplifies how the TJM can handle large-scale open quantum system simulations efficiently even without specialized high-performance hardware. An implementation can be found in the *MQT-YAQS* package available at[52] as part of the *Munich Quantum Toolkit*[53].

**Monte Carlo convergence.** We first examine how the TJM converges with respect to the number of trajectories $N$ and the time step size $\delta t$. As an exact reference, a direct solution of the Lindblad equation via QuTiP[67,68] is used.

In Fig. 4a, the absolute error in the expectation value of a local $X$ operator at the chain's center is plotted, evaluated at $Jt = 1$ for up to $N = 10^4$ trajectories and for several time step sizes $\delta t \in \{0.1, 0.2, 0.5\}$. Each point in the plot represents an average over 1000 independent batches of $N$ trajectories. The dotted lines correspond to a first-order Trotter decomposition of the TJM serving as a baseline. The solid lines correspond to the second-order Trotterization used as a basis for the TJM. The solid black line represents the expected Monte Carlo convergence $\propto 1/\sqrt{N}$ (with prefactor $C = 0.1$).

For the first-order Trotter method (dotted lines), all step sizes induce a plateau, indicating that Trotter errors dominate when $N$ becomes large. In comparison, the second-order TJM approach (solid lines) maintains $\sim C/\sqrt{N}$ scaling for both $\delta t = 0.1$ and $\delta t = 0.2$, while still showing significantly lower error for $\delta t = 0.5$. This confirms that the higher-order Trotterization of the unitary and dissipative evolutions used in the derivation of the TJM reduces the inherent time step error below the level where it competes with Monte Carlo sampling error. While it is possible that for very large $N$ (beyond those shown here) time discretization errors might again appear, in practice our second-order scheme keeps these systematic errors well below the scale relevant to typical simulation tolerances.

**Effect of bond dimension and elapsed time.** Next, we explore how the TJM's accuracy depends on the maximum bond dimension $\chi$ of the trajectory MPS and the total evolution time $T$. We compute the error in a two-site correlator at each bond

$$
\epsilon = |\langle X^{[i]}X^{[i+1]}\rangle - \langle\tilde{X}^{[i]}\tilde{X}^{[i+1]}\rangle|,
$$

where $\langle X^{[i]}X^{[i+1]}\rangle$ is the exact value (via QuTiP) and $\langle\tilde{X}^{[i]}\tilde{X}^{[i+1]}\rangle$ is the TJM result. This is done for each bond at discrete times $Jt \in \{0, 0.1, \ldots, 10\}$ using a time step $\delta t = 0.1$. We vary the bond dimension $\chi \in \{2, 4, 8\}$ and the number of trajectories $N \in \{100, 1000, 10000\}$. The resulting errors, averaged over 1000 batches for each $(N, \chi)$, are shown in Fig. 4b using a color map centered at $\epsilon = 10^{-2}$.

First, we note that the number of trajectories plays a larger role in the error scaling than the bond dimension, however, a

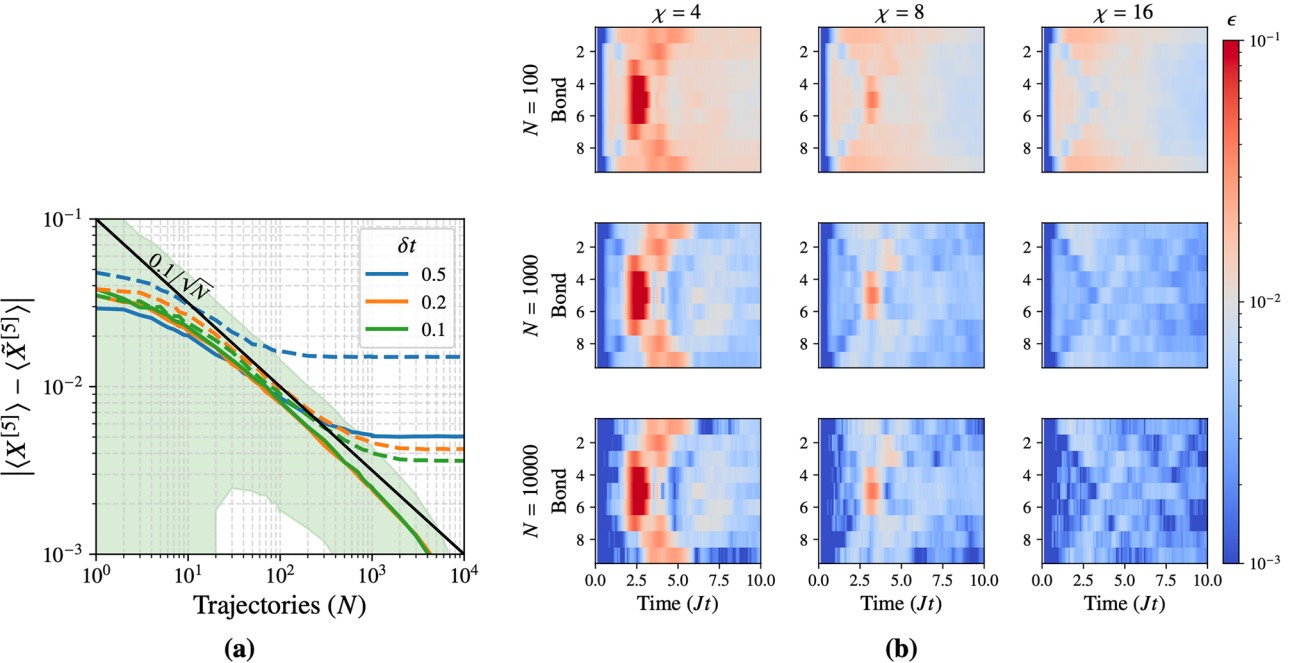

**Fig. 4 | Convergence benchmarks of the TJM. a** Error in the local observable $\langle X^{[5]}\rangle$ at time $Jt = 1$ as a function of the number of trajectories $N$, for several time step sizes $\delta t$. Each point represents the average over 1000 batches. Dotted lines show the corresponding first-order Trotterization results for comparison. The shaded region indicates the standard deviation for $\delta t = 0.1$. The error follows the predicted scaling $\sim C/\sqrt{N}$, with $C \approx 0.1$, demonstrating that the second-order Trotter scheme yields low time-step errors such that $N$ dominates over $\delta t$ in determining convergence.

**b** Convergence of the two-site correlator $\langle X^{[i]}X^{[i+1]}\rangle$ over time $Jt \in [0, 10]$ at fixed $\delta t = 0.1$, shown as a function of trajectory number $N$ and bond dimension $\chi$. Each panel displays the local observable error, averaged over 1000 batches of $N$ trajectories. The colormap is centered at an error threshold $\epsilon = 10^{-2}$: blue indicates lower error, red higher. While increasing $N$ significantly improves global accuracy, a larger bond dimension is still required to resolve localized dynamical features.

higher bond dimension is required for capturing certain parts of the dynamics.

Second, while the simulation time $T$ can affect the complexity of the noisy dynamics, the TJM generally maintains similar accuracy at all times. Indeed, times closer to the initial state ($Jt \approx 0$) exhibit very low errors (blue regions) simply because noise has had less time to build up correlations, and longer times likely show more dissipation which counters entanglement growth.

Finally, the bond dimension $\chi$ plays a comparatively minor role in the overall error. Although $\chi = 4$ sometimes fails to capture certain features (leading to increased error in specific time windows), $\chi = 8$ and $\chi = 16$ give similar results, except for at very specific points in the dynamics such as, in this example, the middle of the chain at around $Jt \approx 3$.

In summary, these benchmarks confirm that (i) the time step error is effectively minimized by the second-order Trotterization, leaving Monte Carlo sampling as the primary source of error, (ii) increasing the number of trajectories $N$ is typically more crucial than increasing the bond dimension $\chi$ for a global convergence, and finally (iii) bond dimension can dominate specific points of the local dynamics.

## New frontiers

In this section, we push the limits of the TJM to large-scale noisy quantum simulations, highlighting its practical utility and the physical insights it can provide. Concretely, we explore a noisy XXX Heisenberg chain described by the Hamiltonian

$$H_0 = -J\left(\sum_{i=1}^{L-1} X^{[i]}X^{[i+1]} + Y^{[i]}Y^{[i+1]} + Z^{[i]}Z^{[i+1]}\right)$$
$$- h\sum_{j=1}^{L} Z^{[j]},$$

at the critical point $J = h = 1$, subject to relaxation ($\gamma_-$) and excitation ($\gamma_+$) noise processes. Each simulation begins with a *domain-wall* initial state

$$|\Psi(0)\rangle = |\sigma_1\sigma_2\ldots\sigma_\ell\ldots\sigma_L\rangle, \quad \sigma_\ell = \begin{cases} 0, & 1 \le \ell < \frac{L}{2}, \\ 1, & \frac{L}{2} \le \ell \le L, \end{cases}$$

such that the top half of the chain is initialized in the spin-down $|0\rangle$ state and the bottom half in the spin-up $|1\rangle$ state, thus forming a sharp "wall". We track the local magnetization $\langle Z\rangle$ at each site as the primary observable of interest.

To demonstrate the scalability of our approach, we simulate system sizes ranging from moderate $L = 30$ to quite large $L = 100$, and then up to $L = 1000$ sites. By examining a wide range of noise strengths and run times, we reveal how noise impacts the evolution of such extended systems-insights that are otherwise out of reach for many conventional methods.

**Comparison with MPO Lindbladians (30 sites).** We benchmark the TJM method against a state-of-the-art MPO Lindbladian solver (implemented via the `LindbladMPO` package[69]) by simulating a 30-site noisy XXX Heisenberg model with a domain-wall initial state localized at site 15. The results are summarized in Fig. 5.

We consider a dissipative scenario with local relaxation and excitation channels $\sigma_\pm$ of equal strength $\gamma = 0.1$, and compare TJM simulations at increasing bond dimension $\chi \in \{2, 4, 8\}$ against a reference MPO simulation with bond dimension $D = 400$. All TJM simulations were performed with only $N = 100$ trajectories, deliberately chosen as a low sampling rate to emphasize the method's practicality for fast, exploratory research. Despite this modest sample size, we observe accurate dynamics, demonstrating that even low-$N$ TJM simulations are capable of producing reliable physical insight. More

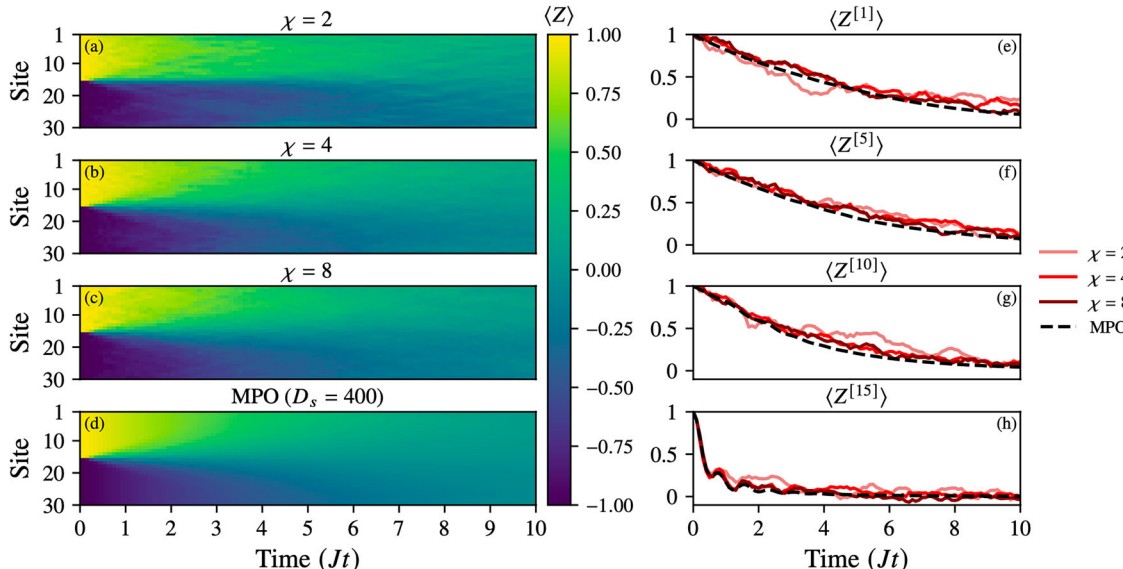

**Fig. 5 | Convergence of the TJM with increasing bond dimension $\chi$ = {2, 4, 8} for a 30-site noisy XXX Heisenberg model with parameters $J = h = 1$ and a domain-wall initial state (wall at site 15).** The noise model includes local relaxation and excitation channels $\sigma_\pm$ with equal strength $\gamma_\pm = 0.1$. **a–d** Site-resolved magnetization $\langle Z^{[i]} \rangle$ over time, compared against a reference simulation using an MPO with bond dimension capped at $D = 400$. **e–h** Time evolution of $\langle Z^{[i]} \rangle$ for selected sites $i = 1, 5, 10, 15$, showing quantitative agreement between TJM and MPO as $\chi$ increases. All simulations use a timestep $\delta t = 0.1$, and TJM is averaged over $N = 100$ trajectories. While the MPO simulation requires over 24 h, the TJM runs complete in under 5 min. This highlights the ability of TJM to efficiently and accurately reproduce both global and local dynamics at modest bond dimension, making it a practical tool for rapid prototyping of noisy quantum dynamics.

precise convergence can be trivially obtained by increasing $N$, making the method tunable in both runtime and accuracy.

Panels (a–d) show the site-resolved magnetization $\langle Z^{[i]} \rangle$ over time, illustrating rapid convergence of the TJM as $\chi$ increases. Even at $\chi = 4$, the TJM closely reproduces the global magnetization profile obtained from the MPO benchmark. All simulations exhibit the expected spreading and dissipation of the initial domain wall under the influence of noise.

Panels (e–h) track the time evolution of $\langle Z^{[i]} \rangle$ at selected sites $i = 1, 5, 10, 15$, revealing excellent agreement between TJM and MPO results, with small stochastic fluctuations at low $\chi$ and early times before the noise model dominates. The consistency across both global and local observables highlights the accuracy of the TJM approach, even under tight computational budgets.

In terms of performance, while the MPO simulation required over 24 h to complete, each TJM simulation (with $N = 100$) ran in under 5 min. This computational efficiency, combined with flexible accuracy control via the trajectory number $N$, makes TJM a powerful tool for rapid prototyping and intuitive exploration of noisy quantum dynamics. We also note that the TJM is positive semi-definite by construction as shown in Appendix 1, while the MPO solver does not guarantee this.

**Validating convergence against analytical results ($L = 100$).** To benchmark our method against known large-scale behavior, we simulate an edge-driven XXX Heisenberg chain of $L = 100$ sites with parameters $J = 1$ and $h = 0$. This model admits an exact steady-state solution in the limit of strong boundary driving[70,71].

Lindblad jump operators act only on the boundary sites,

$$L_1 = \sqrt{\epsilon}\, \sigma_1^+, \qquad L_{100} = \sqrt{\epsilon}\, \sigma_{100}^-,$$

where $\sigma^\pm = \frac{1}{2}(X \pm iY)$. In the long-time limit and for $\epsilon \gg 2\pi/L$, the steady-state local magnetization is given by

$$\langle Z_j \rangle_{\text{ss}} = \cos\left[\pi \frac{j-1}{L-1}\right], \quad j = 1, \ldots, L.$$

We simulate the dynamics using the TJM initialized from the all-zero product state, with parameters $\epsilon = 10\pi$, timestep $\delta t = 0.1$, $N = 100$ trajectories, and bond dimension $\chi = 4$. As shown in Fig. 6, the observed local magnetization $\langle Z_j \rangle(t)$ converges to the analytical profile over time, with absolute deviation

$$\Delta_{\text{steady}}(t) = |\langle Z_j(t) \rangle - \langle Z_j \rangle_{\text{ss}}|$$

falling below $10^{-2}$ for all sites by $Jt \approx 90\,000$. This confirms that the TJM achieves accurate convergence to the correct steady state even at scale and with limited resources.

**Pushing the envelope (1000 sites).** Finally, we demonstrate the scalability of the TJM by simulating a noisy XXX Heisenberg chain of 1000 sites. While our method can handle even larger systems—especially if migrated from consumer-grade to server-grade hardware—we choose 1000 sites here as a reasonable upper bound for our present setup. This test, which took roughly 7.5 h, maintains the same parameters as before: we run a noise-free simulation ($\gamma = 0$) requiring only a single trajectory, and then a noisy simulation with $\gamma = 0.1$, bond dimension $\chi = 4$, and $N = 100$ trajectories at time step $\delta t = 0.5$.

In Fig. 7, we show the time evolution of the noisy and noise-free systems along with the difference $\Delta = \langle Z_{\text{Noisy}} \rangle - \langle Z_{\text{Noise-Free}} \rangle$. While the overall domain-wall structure appears visually similar at a glance, single-site quantum jumps from relaxation and excitation lead to small, localized "scarring" that accumulates into increasingly macroscopic changes by $Jt = 10$. Notably, although the larger lattice provides more possible sites for noise to act upon, the size may confer partial robustness in early stages of evolution. Consequently, we see that even a modest amount of noise ($\gamma = 0.1$) can subtly alter the state in a way that becomes significant over time. These results underscore that the TJM can efficiently capture open-system dynamics in large spin chains on a consumer-grade CPU, thus opening new frontiers for studying the interplay between coherent dynamics and environmental noise at unprecedented scales.

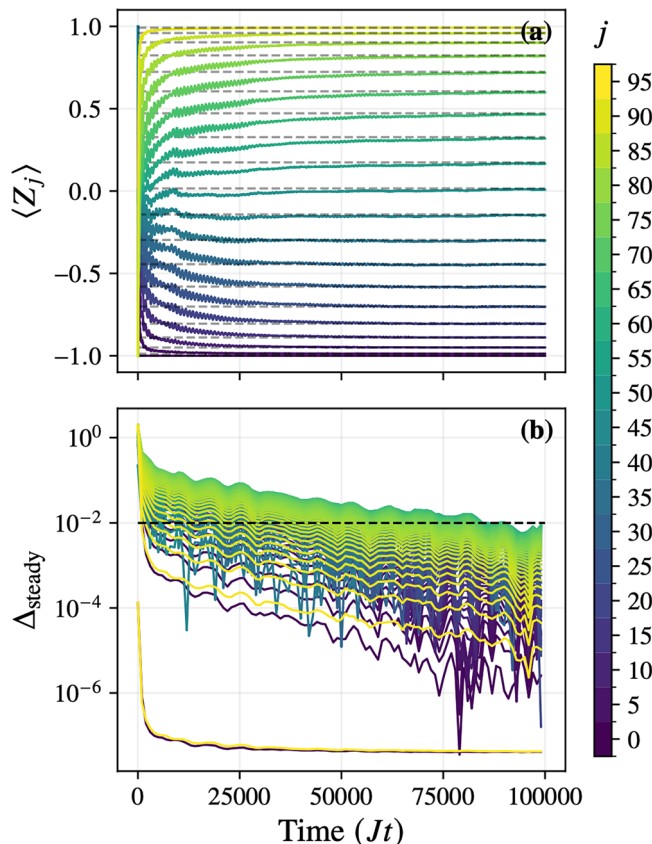

**Fig. 6 | Convergence to analytical steady state of 1000-site noisy XXX Heisenberg chain. a** Time evolution of the local magnetization $\langle Z_j \rangle$ for $j = 1, \ldots, 100$ in an edge-driven XXX chain ($J = 1$, $h = 0$), simulated using the TJM up to $Jt = 100000$ with timestep $\delta t = 0.1$ and boundary driving strength $\epsilon = 10\pi$. Solid lines show the simulated trajectories, and dashed lines indicate the exact steady-state values $\cos[\pi(j-1)/(L-1)]$. **b** Absolute deviation $\Delta_{\text{steady}} = |\langle Z_j \rangle - \langle Z_j \rangle_{\text{ss}}|$ on a logarithmic scale. The horizontal dashed line at $10^{-2}$ marks the target error, which is achieved by all sites by $Jt \approx 90000$.

## Discussion

All quantum system environments are open to some extent, and hence, having powerful tools available that classically simulate interacting open quantum many-body systems is crucial. While recent years have seen a development towards large-scale, state-of-the-art-simulations for quantum ground state and quantum circuit simulations, the same cannot quite be said for the simulation of open quantum systems. This seems a grave omission, given the important role quantum many-body systems play in notions of quantum simulation[4,5]. The present work is meant to close this gap, providing a massively scalable algorithm for the simulation of Markovian open quantum systems by means of tensor networks, paving the way for large-scale classical simulations of open quantum systems matching similar tools for equilibrium problems. By bridging the gap between theoretical frameworks and practical applications, this work not only advances the field of open quantum systems but also contributes to the broader goal of realizing robust and scalable quantum technologies in real-world settings. It is our hope that this work can provide important services in the benchmarking and design of state-of-the-art physical platforms of quantum simulators in the laboratory, as well as inspire and facilitate research into large-scale open quantum systems and noisy quantum hardware that was previously infeasible.

## Methods

This section is devoted to providing the background of key methods and techniques used in this work. We start by presenting core ideas of tensor network methods[2,9,16,72] that this work is building on.

### Matrix product states

Consider a one-dimensional lattice made of $L \in \mathbb{N}$ sites, each corresponding to a local Hilbert space $\mathcal{H}_d$ of dimension $d \in \mathbb{N}$. The Hilbert space of the full lattice is then defined by the iterative tensor product of the $L$ local Hilbert spaces $\mathcal{H} = \bigotimes_{\ell=1}^{L} \mathcal{H}_d$. Elements $|\Psi\rangle$ of this multi-site Hilbert space $\mathcal{H}$ are state vectors defined by

$$|\Psi\rangle = \sum_{\sigma_1, \ldots, \sigma_L = 1}^{d} \Psi_{\sigma_1 \ldots \sigma_L} |\sigma_1, \ldots, \sigma_L\rangle \qquad (62)$$

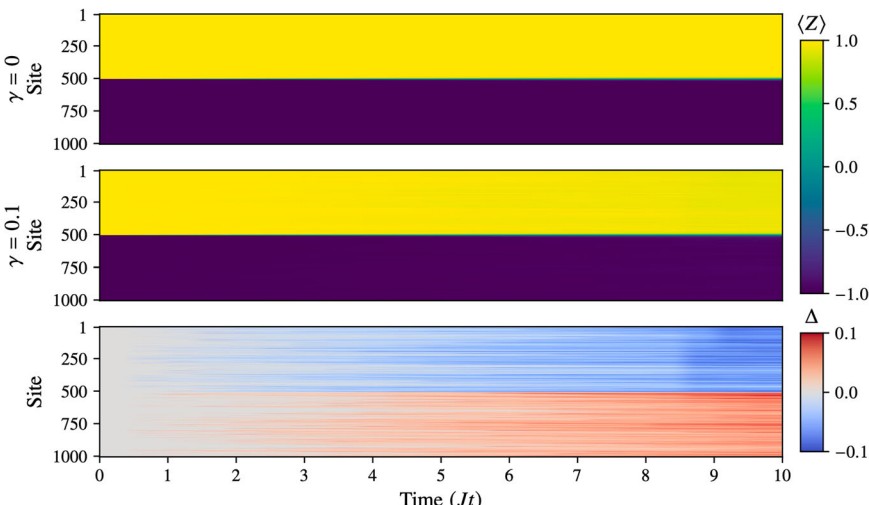

**Fig. 7 | These plots show the results of the evolution of a 1000-site noisy XXX Heisenberg model with parameters $J = 1$, $h = 0.5$ and a domain-wall initial state with wall at site 500.** First, we run the TJM with $\gamma = 0$ to generate a noise-free reference which requires only a single "trajectory". Then, we run it again with $\gamma = 0.1$ using $N = 100$ and bond dimension $\chi = 4$. Next, we run it again with $\gamma = 0.1$ using $N = 100$ and bond dimension $\chi = 4$. Finally, we plot the difference $\Delta$ between these two plots. While visually very similar, we see that the TJM is able to capture even relatively minor noise effects, and, as a result, show how this can eventually lead to macroscopic changes.

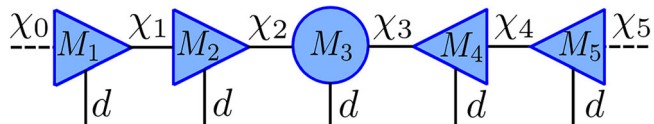

**Fig. 8 | This figure shows a 5-site MPS in mixed canonical form with ortho-gonality center at site 3.** The indices $d_j$ indicate the physical dimensions and $\chi_j$ the bond dimensions. A dashed line indicates a dummy index where $\chi_j = 1$. We use right-pointing triangles to denote the left-canonical form and left-pointing triangles for the right-canonical form. A tensor with no required form is shown by a circle.

where $\sigma_\ell \in [1, \dots, d]$ are the physical dimensions for all $\ell = 1, \dots, L$ and $\Psi \in \mathbb{C}^{d^L}$ with elements $\Psi_{\sigma_1 \dots \sigma_L} \in \mathbb{C}$.

The vector $|\Psi\rangle \in \mathbb{C}^{d^L}$ can be decomposed into a *matrix product state* (MPS)[9,11,73]

$$|\boldsymbol{\Psi}\rangle = \sum_{\sigma_1, \dots, \sigma_L = 1}^{d} M_1^{\sigma_1} \dots M_L^{\sigma_L} |\sigma_1, \dots, \sigma_L\rangle, \tag{63}$$

which is made of $L$ degree-3 tensors

$$M := \{ M_\ell \in \mathbb{C}^{d \times \chi_{\ell-1} \times \chi_\ell} \,|\, \ell = 1, \dots, L \}, \tag{64}$$

consisting of $d$ matrices corresponding to each index $\sigma_\ell$

$$M_\ell := \{ M_\ell^{\sigma_\ell} \in \mathbb{C}^{\chi_{\ell-1} \times \chi_\ell} \,|\, \sigma_\ell = 1, \dots, d \}. \tag{65}$$

This structure's complexity is determined by its bond dimensions $\chi_\ell \in \mathbb{N}$ (where $\chi_0 = \chi_L = 1$), which scale with the entanglement entropy of the quantum state it represents. For the rest of this work we denote a quantum state vector living in $\mathcal{H}$ as $|\Psi\rangle$ and we use the bold notation $|\boldsymbol{\Psi}\rangle$ for the same quantum state vector in an MPS representation.

The MPS representation is non-unique and gauge-invariant for representing a given quantum state such that the individual tensors can be placed in canonical forms which allow many operations to reduce in complexity. These conditions are the left canonical form

$$\sum_{\sigma_\ell = 1}^{d} \sum_{a_{\ell-1} = 1}^{\chi_{\ell-1}} \overline{M}_\ell^{\sigma_\ell, a_{\ell-1}, a_\ell} M_\ell^{\sigma_\ell, a_{\ell-1}, a_\ell} = I \in \mathbb{C}^{\chi_\ell \times \chi_\ell}, \tag{66}$$

and the right canonical form

$$\sum_{\sigma_\ell = 1}^{d} \sum_{a_\ell = 1}^{\chi_\ell} \overline{M}_\ell^{\sigma_\ell, a_{\ell-1}, a_\ell} M_\ell^{\sigma_\ell, a_{\ell-1}, a_\ell} = I \in \mathbb{C}^{\chi_{\ell-1} \times \chi_{\ell-1}}, \tag{67}$$

where $a_\ell \in [1, \dots, \chi_\ell]$ and $\overline{M}$ is the conjugated tensor. Finally, these can be combined to fix the MPS in a mixed canonical form around an orthogonality center at site tensor $j$

$$\sum_{\sigma_\ell = 1}^{d} \sum_{a_{\ell-1} = 1}^{\chi_{\ell-1}} \overline{M}_\ell^{\sigma_\ell, a_{\ell-1}, a_\ell} M_\ell^{\sigma_\ell, a_{\ell-1}, a_\ell} = I \text{ such that } \ell < j,$$
$$\sum_{\sigma_\ell = 1}^{d} \sum_{a_\ell = 1}^{\chi_\ell} \overline{M}_\ell^{\sigma_\ell, a_{\ell-1}, a_\ell} M_\ell^{\sigma_\ell, a_{\ell-1}, a_\ell} = I \text{ such that } \ell > j. \tag{68}$$

A visualization of an MPS in site-canonical form at $j = 3$ can be found in Fig. 8 where the triangular tensors indicate canonical forms and the circular tensor indicates an arbitrary form.

## Matrix product (density) operators

Just as states on a one-dimensional lattice can be represented by MPS, the operators acting upon these states can be represented by a tensor train known as *matrix product operators* (MPO)[74,75]. Suppose we have a bounded operator $O \in \mathcal{B}(\mathcal{H})$ such that

$$O = \sum_{\sigma_1, \sigma_1', \dots, \sigma_L, \sigma_L' = 1}^{d} W^{\sigma_1, \sigma_1', \dots, \sigma_L, \sigma_L'} |\sigma_1, \dots, \sigma_L\rangle \langle \sigma_1', \dots, \sigma_L'|, \tag{69}$$

where $W \in \mathbb{C}^{d^{2L}}$ and for each element $W_{\sigma_1, \sigma_1', \dots, \sigma_L, \sigma_L'}$ we have $\sigma_\ell, \sigma_\ell' \in \{1, \dots, d\}$ for all $\ell = 1, \dots, L$. This coefficient tensor $W$ can then be decomposed into a list of $L$ degree-4 tensors

$$W := \{ W_\ell \in \mathbb{C}^{d \times d \times D_{\ell-1} \times D_\ell} \,|\, \ell = 1, \dots, L \}, \tag{70}$$

created by degree-2 tensors for the indices $\sigma_\ell, \sigma_\ell'$:

$$W_\ell := \{ W_\ell^{\sigma_\ell, \sigma_\ell'} \in \mathbb{C}^{D_{\ell-1} \times D_\ell} \,|\, \sigma_\ell, \sigma_\ell' = 1, \dots, d \}. \tag{71}$$

Next, we define multi indices

$$\boldsymbol{\sigma} = (\sigma_1, \dots, \sigma_L), \quad \boldsymbol{\sigma}' = (\sigma_1', \dots, \sigma_L'),$$

and write

$$|\boldsymbol{\sigma}\rangle \equiv |\sigma_1, \dots, \sigma_L\rangle, \quad \langle\boldsymbol{\sigma}'| \equiv \langle\sigma_1', \dots, \sigma_L'|,$$

as well as

$$W(\boldsymbol{\sigma}, \boldsymbol{\sigma}') := \prod_{\ell=1}^{L} W_\ell^{\sigma_\ell, \sigma_\ell'}.$$

Then the MPO in bold notation is compactly written as

$$\boldsymbol{O} = \sum_{\boldsymbol{\sigma}, \boldsymbol{\sigma}' = 1}^{d} W^{\boldsymbol{\sigma}, \boldsymbol{\sigma}'} |\boldsymbol{\sigma}\rangle \langle\boldsymbol{\sigma}'|. \tag{72}$$

Like the MPS, the computational complexity is determined by the bond dimensions $D_\ell \in \mathbb{N}$, which are related to the operator entanglement[76,77].

Particularly important for this work, density matrices, i.e., mixed quantum states, can be represented in this MPO format[20]. A mixed quantum state $\rho \in \mathcal{B}(\mathcal{H})$ is equivalent to a finite sum of the outer product of $N \in \mathbb{N}$ state vectors $|\Psi_j\rangle$ and weighted by probabilities $p_j$ with $j = 1, \dots, N$ such that

$$\rho = \sum_{j=1}^{N} p_j |\Psi_j\rangle\langle\Psi_j|, \tag{73}$$

and $\sum_j p_j = 1$. If the pure states are represented as MPS according to Eq. (63), this outer product would result in an MPO structure with degree-4 site tensors $W_\ell \in \mathbb{C}^{d \times d \times D_{\ell-1} \times D_\ell}, \ell = 1, \dots, L$ which can be decomposed in a sum of $N$ outer products of MPS given by

$$\{ M_{\ell,j} \in \mathbb{C}^{d \times \chi_{\ell-1} \times \chi_\ell} \,|\, \ell = 1, \dots, L \}_{j=1}^{N} \tag{74}$$

such that each site $W_\ell$ is represented by

$$\sum_{j=1}^{N} \overline{M}_{\ell,j} \otimes M_{\ell,j}. \tag{75}$$

The bond dimensions of the MPO are made of the constituent MPS bonds such that $D_{\ell-1} = \sum_{i=1}^{N} \chi_{\ell-1,i}^2$ and $D_\ell = \sum_{i=1}^{N} \chi_{\ell,i}^2$, where $\chi_{\ell,i}$ is the $\ell$-th bond dimension of the $i$-th MPS. This can be seen in Fig. 9.

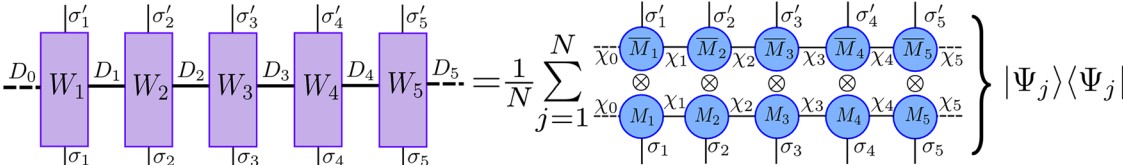

**Fig. 9 | An MPO can be created by a weighted summation of MPS structures and their conjugates as described in section "Matrix product (density) operators".** However, this can lead to massive MPO bond dimensions since the bonds for each state vector $|\Psi_j\rangle$ are combined such that $D_\ell = \prod_{j=1}^N (\chi_\ell^2)_j$. This equivalence is critical for the foundations of this work. However, we avoid creating an MPO directly due to the large bond dimensions.

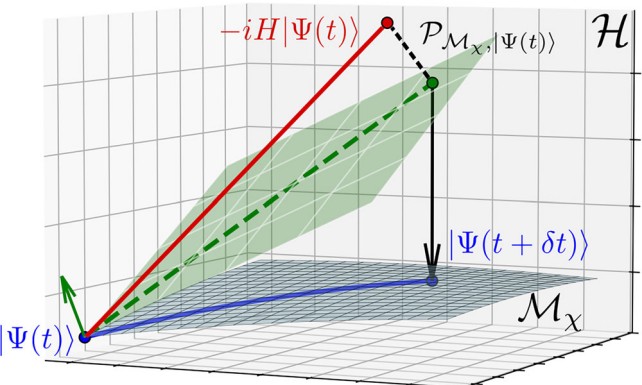

**Fig. 10 | A time-dependent MPS $|\Psi(t)\rangle$ with fixed bond dimensions $\chi$ can be viewed as a point on a manifold $\mathcal{M}_\chi \in \mathcal{H}$.** The time-evolution caused by $-iH|\Psi(t)\rangle$ will generally leave the manifold, requiring a larger bond dimension to represent. TDVP projects this time-evolution to the tangent space of the original manifold (shown by the projector $\mathcal{P}_{\mathcal{M}_\chi, |\Psi(t)\rangle}$). Each point in the tangent space can be described as a linear combination of partial derivatives of the original point such that by solving a set of site-wise coupled differential equations, we can evolve purely on the original manifold and limit the growth in bond dimension to represent $|\Psi(t+\delta t)\rangle$.

## Time-dependent variational principle

**Overview.** Let $|\Psi(t)\rangle \in \mathcal{H}$ for some $t \in \mathbb{R}_+$ be a time-dependent quantum state represented by an MPS with bond dimensions $\chi = \{\chi_0, ..., \chi_L\}$. Then $|\Psi(t)\rangle$ can be understood as an element of a manifold $\mathcal{M}_\chi \subseteq \mathcal{H}$, solely defined by the set of bond dimensions $\chi$. As the bond dimensions increase, this manifold covers a larger part of the Hilbert space such that $\mathcal{M}_\chi \subset \mathcal{M}_{\chi'} \subseteq \mathcal{H}$[66] for $\chi < \chi'$. Generally, time-evolution of some state with the *time-dependent Schrödinger equation* (TDSE) according to some Hamiltonian $H \in \mathcal{B}(\mathcal{H})$ leads to a growth of the bond dimensions, which severely limits the computational efficiency.

By utilizing the manifold picture of MPS, we get a powerful method known as the *time-dependent variational principle* (TDVP). Rather than allowing the bond dimension to grow, this method projects the Hamiltonian to the tangent space of the current MPS manifold before carrying out the time evolution[15] as visualized in Fig. 10. In a more precise form, TDVP solves the projected TDSE

$$\frac{d}{dt}|\Psi(t)\rangle = -iP_{\mathcal{M}_\chi, |\Psi(t)\rangle}H|\Psi(t)\rangle, \qquad (76)$$

where $P_{\mathcal{M}_\chi, |\Psi(t)\rangle} \in \mathcal{B}(\mathcal{H})$ is a projector that projects an MPS onto the tangent space of the manifold $\mathcal{M}_\chi$ at the point $|\Psi(t)\rangle$. TDVP offers significant advantages over other time evolution methods, such as *time-evolving block decimation* (TEBD)[11,45,78], by avoiding Trotter and truncation errors while conserving the system's norm and energy[13,66]. This benefit is transformed into projection error which results in the optimal representation of an MPS at a lower bond dimension (on the smaller manifold) compared to the sub-

optimal truncated MPS[16]. The simplest form of TDVP known as 1TDVP leads to $2L-1$ coupled local *ordinary differential equations* (ODEs) which correspond to a one-site integration scheme similar to the *density matrix renormalization group* (DMRG) method[9]. This can then be extended to an *n*-site integration scheme[15], allowing for the simultaneous integration of neighboring sites. Higher-order *n*TDVP reduces projection error, such that if a Hamiltonian has $n'$-body interactions, no projection error occurs for $n' \le n$, although at the cost of higher computational complexity[16].

**Coupled ordinary differential equations.** For 1TDVP, the projection can be expressed as a projector splitting at each site,

$$P_{\mathcal{M}_\chi, |\Phi\rangle} = \sum_{\ell=1}^L K_{\ell, |\Phi\rangle} - \sum_{\ell=1}^{L-1} G_{\ell, |\Phi\rangle}, \qquad (77)$$

which always depend on the current MPS $|\Phi(t)\rangle$, where

$$K_{\ell, |\Phi\rangle} = \left|\Phi_{\ell-1}^L\right\rangle\left\langle\Phi_{\ell-1}^L\right| \otimes I_\ell \otimes \left|\Phi_{\ell+1}^R\right\rangle\left\langle\Phi_{\ell+1}^R\right|, \qquad (78)$$

represents the Krylov projectors[79] of the MPS $|\Phi\rangle$ that fix the orthogonality center at site $\ell$, and where $G_{\ell, |\Phi\rangle}$ are defined as

$$G_{\ell, |\Phi\rangle} = \left|\Phi_\ell^L\right\rangle\left\langle\Phi_\ell^L\right| \otimes \left|\Phi_{\ell+1}^R\right\rangle\left\langle\Phi_{\ell+1}^R\right|. \qquad (79)$$

We have dropped the time parameter $t$ for ease of notation. Here, the left bipartition of the MPS $|\Phi\rangle$ around the orthogonality center $\ell$ is expressed as

$$\left|\Phi_\ell^L\right\rangle = \sum_{\sigma_1, ..., \sigma_\ell = 1}^d M_1^{\sigma_1} \ldots M_\ell^{\sigma_\ell}|\sigma_1, \ldots, \sigma_\ell\rangle, \qquad (80)$$

and the right bipartition is given by

$$\left|\Phi_\ell^R\right\rangle = \sum_{\sigma_\ell, ..., \sigma_L = 1}^d M_\ell^{\sigma_\ell} \ldots M_L^{\sigma_L}|\sigma_{\ell+1}, \ldots, \sigma_L\rangle, \qquad (81)$$

where $M_\ell$ with $\ell = 1, ..., L$ are the site tensors of $|\Phi\rangle$. When we substitute the definition of $P_{\mathcal{M}_\chi, |\Phi\rangle}$ into Eq. (76), we derive a set of coupled local *ordinary differential equations* (ODEs) that describe the evolution of the state vector $|\Phi\rangle$, where we use the bold notation $\mathbf{H_0}$ to denote the system Hamiltonian in MPO format as

$$\frac{d}{dt}|\Phi\rangle = -i\sum_{\ell=1}^L K_{\ell, |\Phi\rangle}\mathbf{H_0}|\Phi\rangle + i\sum_{\ell=1}^{L-1} G_{\ell, |\Phi\rangle}\mathbf{H_0}|\Phi\rangle. \qquad (82)$$

This is then split into $L$ forward-evolving terms given by

$$\frac{d}{dt}|\Phi\rangle = -iK_{\ell, |\Phi\rangle}\mathbf{H_0}|\Phi\rangle, \qquad (83)$$

and $L - 1$ backward-evolving terms

$$\frac{d}{dt}|\mathbf{\Phi}\rangle = +iG_{\ell,|\mathbf{\Phi}\rangle}\mathbf{H_0}|\mathbf{\Phi}\rangle, \tag{84}$$

**Computational effort of running the simulation**
After having presented the basics of tensor network simulations, we now derive original material on the computational effort of running a simulation with the *tensor jump method (TJM)* as developed in this work. We can analyze the total complexity of the TJM by breaking it down into its constituent components. For this, we have a given number of trajectories $N$ and a total number $n$ of time steps to perform (regardless of the size of $\delta t$ itself). The complexity of calculating $N$ trajectories with $n$ time steps with time step size $\delta t$ depends on the complexity of the dissipative contraction $\mathcal{D}$, the calculation of the probability distribution of the jump operators $L_m$, $m = 1, ..., k$, and the complexity of a one-site TDVP step[80].

The unitary time-evolution $\mathcal{U}$ according to 1TDVP has a complexity of

$$\mathcal{O}(L[d^2D^2\chi_{max}^2 + dD\chi_{max}^3 + d^2\chi_{max}^3]), \tag{85}$$

where $D$ is the maximum bond dimension of the Hamiltonian MPO. Since the majority of time steps is performed with 1TDVP and 2TDVP is not performed with maximum bond dimension, the scaling of 2TDVP is not relevant for the TJM. Additionally, the complexity of 1TDVP and 2TDVP only differs in terms of the local dimension $d$, which scales quadratic in $d$ for 1TDVP and cubic in 2TDVP. Following this, the dissipative sweep $\mathcal{D}$ scales according to

$$\mathcal{O}(Ld^2\chi_{max}^2). \tag{86}$$

The complexity of the stochastic process $\mathcal{J}$ is determined by three parts: the calculation of the probability distribution, the sampling of a jump operator (which is in $\mathcal{O}(1)$ and therefore negligible), and its application. This leads to a total complexity of

$$\mathcal{O}(kd\chi_{max}^3 + d^2\chi_{max}^2), \tag{87}$$

for $k$ total jump operators $L_m$, $m = 1, ..., k$. Putting all this together, we end up with a total complexity of the TJM in

$$\mathcal{O}\Big(Nn\Big[L\Big(d^2D^2\chi_{max}^2 + dD\chi_{max}^3 + d^2\chi_{max}^3 + d^2\chi_{max}^2\Big) + kd\chi_{max}^3 + d^2\chi_{max}^2\Big]\Big),$$

where the dominant terms come from the TDVP sweep and the calculation of the probability distribution. Since the number of jump operators $k$ is related to the system size $L$, e.g., a fixed number of jump operators per site, we can rewrite the complexity using $k = \alpha L$ as

$$\mathcal{O}\Big(NnL\chi_{max}^3[dD + d^2 + \alpha d]\Big), $$

where $\alpha \in \mathbb{N}$. However, since we assume jumps only occur rarely, the TDVP sweep dominates the stochastic process such that this reduces to

$$\mathcal{O}\Big(NnL\chi_{max}^3[dD + d^2]\Big), \tag{88}$$

where we assume $d, D \ll \chi_{max}$.

In comparison, the runtime to solve the Lindblad equation directly scales as $\mathcal{O}(nd^{6L})$ when using the superoperator formalism[55].

The MCWF method, with the same assumptions made for the TJM, requires $\mathcal{O}(Nnd^{3L})$ and a Lindblad MPO requires $\mathcal{O}(nLd^4D_H^2D_s^2)$ such that $D_s$ is the bond dimension of the MPO representing the density matrix (where we expect $D_s \gg \chi_{max}$), and $D_H$ is the bond dimension of the Hamiltonian MPO format using the $W^{II}$ algorithm (analogous to $\chi_{max}$ and $D$ in the TJM, respectively)[81].

**Resources required for storing the results**
For a given maximum bond dimension $\chi_{max}$, the memory complexity to store $N$ MPS trajectories is given by $\mathcal{O}(NLd\chi_{max}^2)$. Exactly solving the Lindblad equation in matrix format requires storing the density matrix and all other operators in its full dimension. When stored as a complex-valued matrix, $\rho$ has a memory complexity $\mathcal{O}(d^{2L})$ such that it scales exponentially for increasing $L$. Storing $N$ trajectories $|\Psi\rangle \in \mathbb{C}^{d^L}$ reduces the memory complexity from $\mathcal{O}(d^{2L})$ to $\mathcal{O}(Nd^L)$ compared to the Lindbladian approach. Meanwhile, an MPO simulation requires $\mathcal{O}(Ld^2D_s^2)$ to store the time-evolved density matrix.

We can compare the TJM and the MCWF complexities to determine the $\chi_{max}$ necessary for the TJM to be more compact. This results in the requirement that

$$\chi_{max} < \sqrt{\frac{d^L}{Ld}}. \tag{89}$$

Since $\sqrt{d^L/(Ld)} \to \infty$ as $L \to \infty$, we see that for large system sizes the TJM is always more compact than the MCWF. When compared against the MPO, we similarly see that the TJM is more compact if

$$\chi_{max} < D_s\sqrt{\frac{d}{N}}, \tag{90}$$

where we expect $D_s \gg \chi_{max}$ for large-scale applications, particularly those with long timescales.

**Resources required for calculating expectation values**
Naturally, we can also use the stored states to solve expectation values. If we sample a local observable $O \in \mathcal{B}(\mathcal{H})$ at a time step $t$, each method has distinct requirements to calculate $\langle O(t)\rangle$. The MPS trajectories from the TJM can be used to calculate the expectation value of an MPO by performing $N$ MPS-MPO-MPS contractions. This results in a complexity of $\mathcal{O}(NL(dD\chi_{max}^3 + d^2D^2\chi_{max}^2))$ for an MPO with max bond dimension $D$[16,82]. With the assumption that $\chi_{max} > d^2, D^2$ in most cases this is dominated by the cubic term, resulting in complexity

$$\mathcal{O}(NLdD\chi_{max}^3). \tag{91}$$

In comparison, using the density matrix $\rho(t)$ from the Lindbladian, calculating $\langle O(t)\rangle = \text{Tr}[\rho(t)O]$ has complexity $\mathcal{O}(d^{6L})$. Replacing the density matrix with the trajectories of the MCWF reduces this to $\mathcal{O}(Nd^{4L})$ while an MPO requires $\mathcal{O}(Ld^2D_s^3)$.

## Data availability
The data that support the findings of this study are available from the corresponding author upon reasonable request.

## Code availability
The source code used in this study is available open-source as part of the Munich Quantum Toolkit's Yet Another Quantum Simulator (MQT YAQS).

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

## Acknowledgements

We would like to thank Phillipp Trunschke for useful discussions. This work has been supported by the Munich Quantum Valley, which is supported by the Bavarian state government with funds from the Hightech Agenda Bayern Plus, and for which this work reports on results of a joint-node collaboration involving the FU Berlin and two teams at the TU Munich. It has also been funded by the the European Union's Horizon 2020 Quantum Flagship innovation program Millenion (grant agreement No. 101114305) for which this again constitutes joint node work by the FU Berlin and the TU Munich, and the Einstein Research Unit on Quantum Devices (for which this work reflects once more joint work now involving the Weierstrass Institute, the Zuse Institute and the FU Berlin). The team at Technische Universität München has received additional funding from the European Research Council (ERC) under the European Union's Horizon 2020 research and innovation program (grant agreement No. 101001318). The team at the Freie Universität Berlin has been additionally supported by the DFG (CRC 183, FOR 2724), the BMFTR (MuniQC-Atoms), the European Union's Horizon 2020 Quantum Flagship innovation program PasQuans2 (grant agreement No. 101113690), Berlin Quantum, and the European Research Council (ERC DebuQC).

## Author contributions

A.S. conceived the project, developed the method, and performed the simulations. M.F., M.E., and M.H. contributed to the mathematical analysis and convergence proofs. P.G. and J.E. provided substantial support in defining results to be shown, in writing the manuscript, and in presentation. R.M.M., R.W., and C.B.M. supervised the work, offered theoretical support in development of the method, and contributed to manuscript preparation. All authors discussed the results and contributed to the final manuscript.

## Funding

## Competing interests

The authors declare no competing interests.
