## [Transparent Peer Review file · Nature Communications]

Large-scale stochastic simulation of open quantum systems

Corresponding Author: Professor Jens Eisert

Version 0:

Reviewer comments:

Reviewer #1

(Remarks to the Author)

This revised version of the manuscript is greatly improved and is now suitable for publication. The authors have satisfactorily answered all my questions. The text is well-written, with a comprehensive explanation of the "Tensor Jump Method" and a valuable exploration of its limits. It fits the scope of Nature Communications as a regular article.

Reviewer #2

(Remarks to the Author)

In the revised version the authors have answered my technical concerns about their implementation of TDVP. As I indicated previously, the development of the sampling MPS and more broadly the implementation of a non-hermitian operator within the time-dependent variational principle with reduced timestep errors are new. In my view these developments are definitely of interest to specialists working on MCWF with MPS.

However, despite several reviewer comments including mine, the authors still substantially understate the position of previous literature also in the revised manuscript, and overstate the importance of timestepping errors in previous calculations relative to MPS truncation errors. I don't believe that the reader gets a clear understanding of the relative importance of the developments here, despite the additional text on the second page.

To be specific, in the abstract/introduction the authors write: "In this work, we introduce the tensor jump method (TJM), a scalable, embarrassingly parallel algorithm for stochastically simulating large-scale open quantum systems, specifically Markovian dynamics captured by Lindbladians". This method is built on three core principles where, in particular, we extend the Monte Carlo wave function (MCWF) method to matrix product states, use a dynamic time-dependent variational principle (TDVP) to significantly reduce errors during time evolution, and introduce what we call a sampling MPS to drastically reduce the dependence on the simulation's time step size."

As noted now more clearly by the authors in the revised version on the second page, previous work combined MCWF with matrix product states (MPS), and made use of TDVP. In the present manuscript the authors find a more stable implementation of TDVP for non-hermitian systems. I don't think this is properly conveyed in the text quoted above.

The authors then write: "We demonstrate that this method scales more effectively than previous methods". I don't think that this is accurate for the method as a whole for scenarios that most previous implementations of MCWF+MPS methods have examined. The errors from timestepping will scale better here, but most calculations in the past were bottlenecked by truncation errors due to restricted bond dimensions in the MPS. This is also my main concern in the authors' comparison with literature on page 2 – they conclude that they provide "principled solutions to long-standing numerical bottlenecks", but in most cases the main bottleneck is the one they do not address. I certainly don't mean that this work is not useful – on the contrary, as I've written before, I think that this is a nice improvement within the question of how to do timestepping, and the authors have shown that it can become central in some regimes. But I think that the impact of this development is overstated in the manuscript, and that the work and the change it provides are not put properly in the context of previous literature.

Finally, and as a more minor point, the authors identify the regime of strong dissipation (and long times) as the main area of

application of their work. In this limit, I wonder about the error scaling with timestep if they retain a 1st order treatment of dissipative terms (this was a bottleneck that needed to be solved in early calculations with MCWF when dissipation was strong). The authors discussed this in their response, and indicated that their focus was on improving the scaling of the Hamiltonian part with timestep. But this is something to be particularly careful of in the strong dissipation limit, and it would be good to understand when errors from this become dominant.

With a properly reworked introduction, I believe that this would be a good paper for a specialised journal, but I do not recommend publication in Nature Communications.

Reviewer #3

(Remarks to the Author)

I thank the authors for addressing the comments that I had previously pointed out. I am happy to see that the authors indeed took the referees comments seriously and put solid effort to change the manuscript. Most of my comments were appropriately addressed. In my original concern about changing the physical observable to better test the performance of TJM in terms of its dependence on bond dimension I suggested pairwise entanglement (quantified by Wootters' concurrence). While the authors changed to test the XX bond instead, the spirit remains the same, as bond operators somehow contain certain correlations making them sensitive to the bond dimension as well. I agree that the paper has been greatly improved and it is suitable for Nature Communications.

Reply to Reviewer #2

We sincerely appreciate the comments of the referee. Of course, we are well aware that we are standing on the shoulders of giants, while at the same time improving upon a body of literature that provides insights into the study of open quantum systems using tensor networks. At the same time, we improve previously known methods and overcome established bottlenecks for physically highly relevant scenarios in a substantial manner. We have explained precisely how the errors from time stepping scale better in our approach, which we believe has an important impact on how we think about simulating open quantum systems using classical methods. To be cautious, we have altered the wording from “the long-standing numerical bottlenecks” to “some of the long-standing numerical bottlenecks.” Otherwise, we are unaware of any additional literature that we should cite and discuss and hope we have fairly given credit to those who have contributed to those ideas.

Then, we would like to politely point out that we do not claim that the method is primarily for strong dissipation, but that it also works well in this regime. We have found the way to treat the order of expressions as we do it as a sweet spot that works well, and the error scaling is discussed in Results.C.3, where also rigorous guarantees are being presented. We share the curious attitude of the referee and would like to explore more regimes of extreme dissipation in future work.